# Substrate transport and anion permeation proceed through distinct pathways in glutamate transporters

**Mary Hongying Cheng[1†], Delany Torres-Salazar[2†], Aneysis D Gonzalez-Suarez[2], Susan G Amara[2], Ivet Bahar[1]\***

[1]Department of Computational and Systems Biology, School of Medicine, University of Pittsburgh, Pittsburgh, United States; [2]Laboratory of Molecular and Cellular Neurobiology, National Institute of Mental Health, National Institutes of Health, Bethesda, United States

**Abstract** Advances in structure-function analyses and computational biology have enabled a deeper understanding of how excitatory amino acid transporters (EAATs) mediate chloride permeation and substrate transport. However, the mechanism of structural coupling between these functions remains to be established. Using a combination of molecular modeling, substituted cysteine accessibility, electrophysiology and glutamate uptake assays, we identified a chloride-channeling conformer, *iChS*, transiently accessible as EAAT1 reconfigures from substrate/ion-loaded into a substrate-releasing conformer. Opening of the anion permeation path in this *iChS* is controlled by the elevator-like movement of the substrate-binding core, along with its wall that simultaneously lines the anion permeation path (*global*); and repacking of a cluster of hydrophobic residues near the extracellular vestibule (*local*). Moreover, our results demonstrate that stabilization of *iChS* by chemical modifications favors anion channeling at the expense of substrate transport, suggesting a mutually exclusive regulation mediated by the movement of the flexible wall lining the two regions.

*For correspondence: bahar@pitt. edu

[†]These authors contributed equally to this work

**Competing interests:** The authors declare that no competing interests exist.

## Introduction

Excitatory amino acid transporters (EAATs) regulate glutamatergic signaling by mediating the clearance of extracellular (EC) glutamate from the synapse into neuronal and glial cells (*Danbolt, 2001*). EAATs accomplish this task through a secondary active transport mechanism that couples the influx of three sodium ions and one proton and the efflux of one potassium ion to the inward translocation of one glutamate molecule (*Levy et al., 1998*; *Zerangue and Kavanaugh, 1996*). In addition, they mediate a thermodynamically uncoupled substrate-gated chloride conductance (*Fairman et al., 1995*; *Wadiche et al., 1995*), which modulates cell excitability serving as a sensor of EC glutamate concentrations (*Melzer et al., 2005*; *Veruki et al., 2006*; *Wersinger et al., 2006*).

Crystallographic data obtained from the archaeal orthologs $Glt_{Tk}$ from *Thermococcus kodakarensis* (*Guskov et al., 2016*; *Jensen et al., 2013*) and $Glt_{Ph}$ from *Pyrococcus horikoshii* in different conformational states (reviewed by Drew and Boudker [*Drew and Boudker, 2016*]), provide a framework for establishing our current understanding of structure-function relationships in mammalian EAATs. In adition, functional studies (for review see [*Fahlke et al., 2016*; *Jiang and Amara, 2011*]) and structure-based simulations (*Bahar, 2014*; *Crisman et al., 2009*; *DeChancie et al., 2011*; *Gu et al., 2009*; *Heinzelmann and Kuyucak, 2014a*, *2014b*; *Huang and Tajkhorshid, 2008*; *Jiang et al., 2011*; *Lezon and Bahar, 2012*; *Li et al., 2013*; *Machtens et al., 2015*; *Shrivastava et al., 2008*; *Stolzenberg et al., 2012*; *Vergara-Jaque et al., 2015*; *Zomot and Bahar,*

*2013*) have substantially improved our understanding of the molecular mechanism of substrate transport and anion permeation. EAATs display an evolutionarily conserved trimeric quaternary structure (*Gendreau et al., 2004*; *Haugeto et al., 1996*; *Yernool et al., 2003*, *2004*). Each protomer contains eight transmembrane (TM) helices and two reentrant hairpin loops, HP1 and HP2 (*Yernool et al., 2004*). TM1, TM2, TM4 and TM5 make intersubunit contacts and form the trimerization domain (*Boudker et al., 2007*; *Lezon and Bahar, 2012*; *Reyes et al., 2009*; *Yernool et al., 2004*). A scaffold formed by these domains surrounds the highly conserved core (of each subunit) composed of HP1, HP2, TM7 and TM8. These core elements, together with TM3 and TM6, form the transport domain that moves in an elevator-like motion across the membrane to enable substrate translocation (*Lezon and Bahar, 2012*; *Reyes et al., 2009*).

Glt$_{Ph}$ has been crystallized in two end-states: outward-facing state (OFS) (*Boudker et al., 2007*; *Yernool et al., 2004*) and inward-facing state (IFS) (*Reyes et al., 2009*). Moreover, Verdon and Boudker identified an asymmetric intermediate state where one of the protomers was in an intermediate outward-facing state (*i*OFS) and two others in IFS (*Verdon and Boudker, 2012*) (*Figure 1A*). Strikingly, this crystal structure closely resembles our earlier prediction (*Jiang et al., 2011*), which suggested that Glt$_{Ph}$ is likely to sample intermediate conformations, with 'mixed' states of the subunits. Our study further showed that these conformations are likely to be on-pathway intermediates sampled during transitions between the OFS and IFS (*Jiang et al., 2011*).

An initial effort to define regions in EAATs involved in anion flux identified residues in TM2 (*Ryan et al., 2004*) that, when mutated, affected anion conduction, implicating this region in anion permeation. Additional residues in TM2 (*Kovermann et al., 2010*; *Ryan and Mindell, 2007*; *Ryan et al., 2004*; *Winter et al., 2012*), TM5, HP1 and TM7 (*Cater et al., 2014*; *Huang and Vandenberg, 2007*; *Machtens et al., 2015*; *Torres-Salazar et al., 2015*) also appear to contribute to chloride permeation and/or anion channel gating. The asymmetric Glt$_{Ph}$ structure that was captured in the *i*OFS provided evidence for an aqueous accessible cavity that was also proposed as a potential anion permeation pathway (*Verdon and Boudker, 2012*). EAAT1 residues in this cavity were tested by *Cater et al. (2014)*, who suggested that the interface between the transport and trimerization

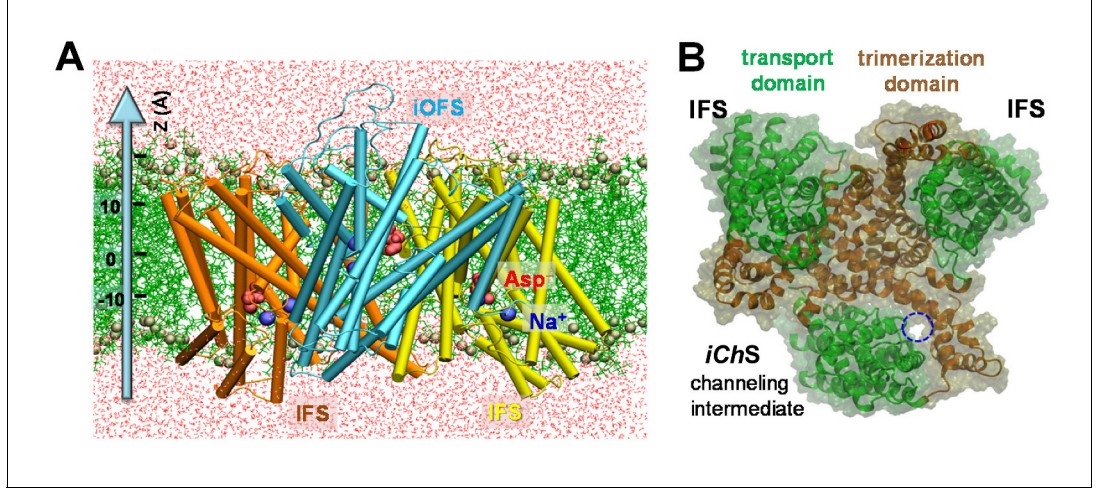

**Figure 1.** Structure of archaeal transporter Glt$_{Ph}$ and computationally predicted anion-conducting channel. (**A**) Structure of Glt$_{Ph}$ that captured one of the protomers in an *i*OFS. The structure is embedded here into POPC membrane bilayer and solvated by 0.1 M NaCl solution (not shown). The substrates (aspartate) and co-transported Na$^+$ ions resolved in the crystal structure are shown in *red* space-*filling* representation and *blue spheres*, respectively. POPC lipids are represented by *green* lines with their phosphorus atoms shown in *tan spheres*. Water molecules are shown in *red dots*. Z-axis directs from the intracellular to extracellular solution. (**B**) Top view of the Glt$_{Ph}$ trimer with one subunit in the anion-channeling state (*iCh*S). A continuous channel (*blue* dashed circle) was intermittently formed as the *i*OFS protomer reconfigured toward the IF state.

The following figure supplement is available for figure 1:

**Figure supplement 1.** The putative anion pore proposed in the *i*OFS (*Verdon and Boudker, 2012*) shows a constriction zone near S65, Y195, M286 and P304.

domains could facilitate anion permeation (*Cater et al., 2014*). Recent molecular dynamics (MD) simulations by Fahlke's group (*Machtens et al., 2015*) showed that the formation of anion-selective permeation pathway is unlikely in the *i*OFS proposed earlier (*Verdon and Boudker, 2012*), but required a global rearrangement of the *i*OFS that included a lateral movement of the transport domain to facilitate the anion pore opening. Data obtained using fluorescence spectroscopy in Glt$_{Ph}$ and electrophysiological recordings in mammalian EAATs also identified several residues at the interface between the transport and trimerization domains that appear to directly interact with chloride, and when mutated, alter the anion-to-cation selectivity and unitary anion conductance (*Machtens et al., 2015*).

Although these studies broadly agree that anion permeation takes place between the transport and trimerization domains, they differ in the conformational states that mediate anion conduction and the precise residues involved. Here, we use a different set of approaches to characterize the structural features of the anion conducting state, to distinguish the residues lining the pore and to resolve the structural elements involved in the coupling between anion conduction and substrate transport. First, using a combination of advanced computations and experiments, we identified a *chloride-channeling intermediate state*, *i*Ch*S* (*Figure 1B*). The *i*Ch*S* is transiently visited during the passage between these *i*OFS and IFS, consistently observed in both Glt$_{Ph}$ and EAAT1. Our simulations clearly showed how the opening of a continuous anion channel is facilitated by the movement of a *flexible wall* between the substrate-binding cavity and the anion conduction pathway, whereas in the *i*OFS and IFS, the continuity of the channel was compromised by constricted areas. We further identified a cluster of hydrophobic residues (F50, V51 and L212 in Glt$_{Ph}$; L88, M89 and L296 in EAAT1) that line the permeation pathway. We probed the accessibility of residues predicted by our simulations to line the anion permeation pore in the *i*Ch*S* using the substituted cysteine accessibility method (SCAM) (*Karlin and Akabas, 1998*), shown earlier to successfully identify pore-forming residues in many ion channels and transporters (*Contreras et al., 2010*; *Fahlke et al., 1997*; *Holmgren et al., 1996*; *Lin and Chen, 2003*; *Pascual et al., 1995*; *Reyes and Gadsby, 2006*; *Takeuchi et al., 2008*). Different engineered cysteine substitutions were introduced into a fully functional EAAT1 that lacks the three endogenous cysteine residues (EAAT1 Csls WT) (*Seal et al., 2001*; *Torres-Salazar et al., 2015*). Each individual mutant was expressed in *Xenopus* oocytes and incubated with different MTS (methanethiosulphonate) reagents. We combined electrophysiological recordings and radiolabeled glutamate uptake assays to monitor residue accessibility and the effect of the modification on anion permeation and substrate transport. Among the residues predicted by our simulations to be part of the proposed anion-channeling pathway, two extracellularly exposed residues, M89 in TM2 and L296 in TM5 reacted rapidly to MTS reagents and reduced anion permeation without affecting substrate transport. In addition, we observed that the accessibility of these residues was reduced or eliminated under conditions that favor OFS, *i*OFS or IFS, which our simulations predict as non-conducting states, suggesting that exposure of these residues to thiol-modifying reagents is state-dependent.

Our results confirm that the anion-selective pathway is conserved in archaeal and mammalian EAATs and that it consists of a cavity between the transport and the trimerization domains (*Cater et al., 2014*; *Machtens et al., 2015*) that opens up upon structural rearrangements from *i*OFS out of the transport cycle (*Machtens et al., 2015*). Moreover, our work provides novel evidence that the anion pore opening consists of a cavity that is distinct from the substrate translocation path and suggest that both pathways are limited by a flexible wall, whose movement dictate whether that monomer exclusively facilitates anion permeation or substrate translocation.

## Results

### A water channel intermittently forms during transition of the substrate-loaded subunit from the *i*OFS to the IFS

We first examined the putative anion permeation pathway in the *i*OFS (*Figure 1A*) suggested by Verdon and Boudker (*Verdon and Boudker, 2012*). We observed that the particular path was blocked by tight interactions near M286, Y195, S65 and P304 (*Figure 1—figure supplement 1*). The pore radius near this constriction zone was 0.9 ± 0.4 Å, which would render the pore impermeable to chloride ions (1.8 Å in radius). On the other hand, M286 (in HP1) is separated from Y195 by over

15 Å in the IFS, which might easily accommodate an anion passage. We then hypothesized that the blockage due to the tight interaction of M286 with Y195 in the *i*OFS might be alleviated as the subunit undergoes a conformational transition from *i*OFS to IFS. This motivated a series of runs (*Supplementary file 1*) using the approach adopted in earlier studies (*Cheng and Bahar, 2013*, *2014*).

*Video 1* illustrates the change in inter-residue interactions as the subunit reconfigures from *i*OFS to IFS. During these runs, we did not observe continuous water occupancy in the region suggested by Verdon and Boudker. While the intracellular (IC) portion of that region was hydrated up to I61, the EC portion near V58 remained dehydrated, ruling against the possibility of anion channeling. Instead, a water channel transiently formed in another (nearby) region at the interface between the transport and trimerization domains (encircled in *Figure 1B*), during the transition. We refer to this new in silico resolved conformer as the *intermediate channeling state (iChS)*. The hydrated channel, was invariably observed in the presence of either two or three bound Na$^+$ ions and the substrate (aspartate in Glt$_{Ph}$) (*Figure 2* and *Figure 2—figure supplement 1*).

## The subunit in the *i*ChS becomes fully hydrated and further opens to expose a hydrophobic cluster during unbiased simulations

The *i*ChS first showed a minimum pore radius of 1.6 ± 0.2 Å, which is not wide enough to allow for chloride permeation; but further equilibration with conventional MD (cMD) simulations in the presence of externally applied voltages (see Materials and methods) led to *full occupancy* of the channel by *continuous* water. The increase in water occupancy in the presence of an electric field may reflect the voltage-gating characteristics of the channel (*Machtens et al., 2015*; *Melzer et al., 2003*, *2005*). Pore lining residues (*Figure 2A*) generally agree with those reported by Fahlke's group (*Machtens et al., 2015*). In our simulations, the cluster of hydrophobic residues F50, V51 and L212 underwent a significant reconfiguration, to give rise to a pore opening in the *i*ChS (*Figure 2B*). In the *i*OFS and IFS, this particular region was blocked by tight packing of the same residues. The opening of the water channel was enabled by the concerted downward movement of the two helical hairpins HP1 and HP2, together with the rotational isomerization of F50, V51 and L212 side chains (*Figure 2B*). The minimum pore radius fluctuated around 2.3 ± 0.3 Å during the simulations, which is comparable to the value of 2.4 Å reported for the crystal structure of an open glutamate-gated chloride channel (*Hibbs and Gouaux, 2011*). Whereas our minimum pore radius was calculated in the absence of chloride, the occupancy of chloride was observed to enlarge the pore up to 2.6 ± 0.2 Å, showing certain flexibility at this constriction to attain the experimentally-determined pore radii of >2.5 Å (*Wadiche and Kavanaugh, 1998*). The chloride ion was detected to translocate across this channel in one of the runs, indicating that *i*ChS conformer presents a pore permeable to chloride.

We then estimated the solvent-accessible surface area (SASA) for residues predicted by our simulations to line the water accessible and chloride permeable pathway in different conformational states. On the EC side, V51, L212, F50 and V209 showed a significantly higher SASA value in the *i*ChS than in either the *i*OFS or the IFS (*Figure 3A*). In particular, F50 and V209 were almost completely buried in the IFS or the *i*OFS,

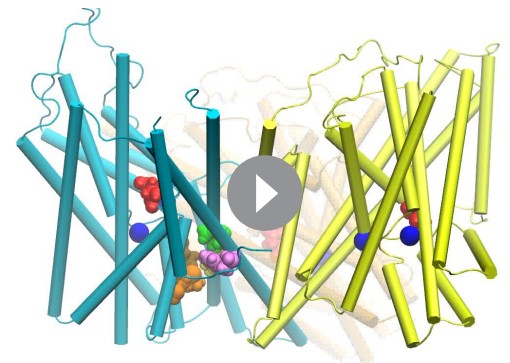

**Video 1.** Transition of the Glt$_{Ph}$ subunit in the *i*OFS (*blue*) into IFS during the global motion of the trimeric transporter, observed in tMD simulations. The two other subunits (shown in *yellow*, and *orange* in the background) are in the IFS. The transporter is in substrate-loaded state (all subunits have a bound aspartate, shown in *red space-filling*). Each subunit also has two bound sodium ions (*blue spheres*). Residues I61, S65, Y195, M286 and P304 are shown *green*, *yellow*, *pink*, *orange*, and *tan* space-filling representations, in the reconfiguring subunit. M286 (*orange*) and P304 (*tan*) get away from S65 and Y195, to induce an opening in the IC-facing end of the cavity, although the putative channeling pore remains closed in the *i*OFS.

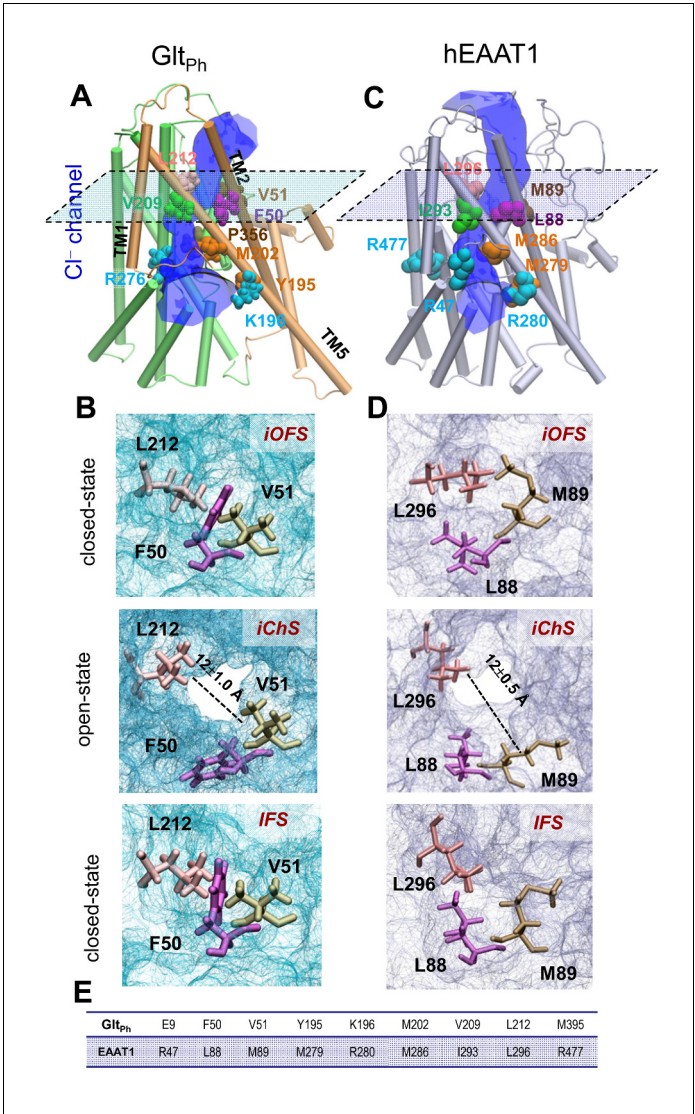

**Figure 2.** Opening of a channel upon transition of the transporter from *i*OFS to inward-facing state (IFS) observed for Glt$_{Ph}$ and EAAT1. Side view of the channeling intermediate *iCh*S in (**A**) Glt$_{Ph}$ and (**C**) EAAT1. The channel (*blue*) is at the interface between the transport (*green*) and trimerization (*brown*) domains. (**B and D**) Representation of pore forming residues in the *i*OFS (*upper panel*: closed-channel state), the *iCh*S (*middle panel*: open-channel state) and the IFS (*lower panel*: closed-channel state) of (**B**) Glt$_{Ph}$ and (**D**) EAAT1. Note that the center-of-mass distance between L212 and V51 is increased from $7.5 \pm 0.5$ Å (closed-channel state) to $12.0 \pm 1.0$ Å (open-channel state) in Glt$_{Ph}$; and that between L296 and M89 is increased from $7.0 \pm 0.5$ Å (closed-channel state) to $12.5 \pm 1.0$ Å (open-channel state) in EAAT1. (**E**) List of selected orthologous residues between EAAT1 and Glt$_{Ph}$.

The following figure supplement is available for figure 2:

**Figure supplement 1.** The hydrated channel was similarly observed in the presence of either two or three bound Na$^+$ ions together with the substrate (aspartate in Glt$_{Ph}$).

and became solvent accessible only upon transition to the *iCh*S. On the IC side, Y195 showed a higher accessibility in the *iCh*S when compared to IFS or *i*OFS. These values provide additional evidence supporting that these residues may be forming part of the anion permeation pathway.

Using our equilibrated *iCh*S as template, we constructed a homology model for EAAT1 and the model was subjected to multiple cMD runs of 100 ns (***Supplementary file 1***). The results presented in ***Figures 2C–D and*** ***3B*** corroborated the behavior observed for Glt$_{Ph}$: mainly, the same region,

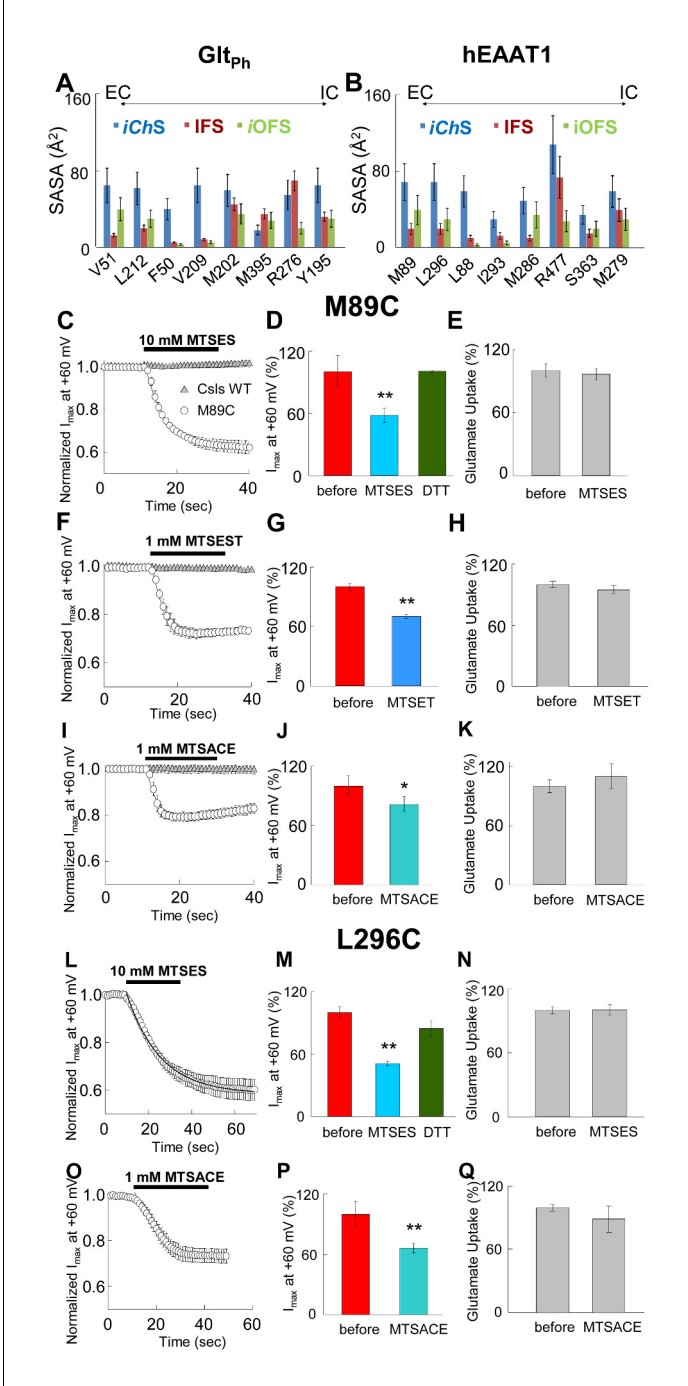

**Figure 3.** Accessibility of EAAT1 pore-forming residues to solvent and MTS-reagents. (**A–B**) Representation of residue solvent-accessible surface area (SASA) for several residues predicted by our simulations to line the water accessible and chloride permeable pathway in the different conformational states: intermediate channeling (*iChS*, *blue*), inward-facing (IFS, *dark red*) and intermediate outward-facing (*iOFS*, *green*). The SASA was estimated for Glt$_{Ph}$ (**A**) and hEAAT1 (**B**). (**C–Q**) The *left* panels represent the averaged current amplitudes obtained at a constant pulse at +60 mV to monitor modification rates of the different MTS-reagents used ($n > 4$). The bars in the center panels show the current amplitude at +60 mV before (*red bars*) and after (different *blue bars*) 3 min of the application of the MTS-reagents ($n > 6$). For the case of MTSES (**D and M**), the current amplitude was measured after application of 1 mM DTT following the application of 10 mM MTSES (*green bars; n > 3*). The *right* panels show a bar graph representation of the radiolabeled glutamate uptake before and 3 min after the application of the reagents at saturating concentrations of glutamate (200 nM radiolabeled Glu⁻ and 500 μM cold Glu⁻) ($n > 20$).

*Figure 3 continued on next page*

*Figure 3 continued*

Panels **C**) to **K**) represent oocytes expressing M89C, while **L**) to **Q**) represent oocytes expressing L296C. For panels **C**), (**F**) and **I**), the application of the MTS-reagents is compared in oocytes expressing M89C (*open circles*) and cysteineless (Csls) WT (*grey triangles*, n > 3). All modification experiments were done in at least two batches of oocytes.

The following figure supplement is available for figure 3:

**Figure supplement 1.** Application of 10 mM MTSES decreases anion current without affecting substrate transport.

lined by the counterparts of Glt$_{Ph}$ residues became continually hydrated. The key roles of the Glt$_{Ph}$ hydrophobic residues F50, V51 and L212 at the pore constriction zone, were assumed by their EAAT1 counterparts, L88, M89 and L296, and were indicated by significant increase in the SASA of these residues in the *iCh*S (*Figure 3B*).

## SCAM and electrophysiological recordings confirm the exposure of hydrophobic residues upon anion channel opening in the *iCh*S of EAAT1

To probe the anion permeation across the fully hydrated pore revealed by our simulations, we created individual cysteine substitutions for human EAAT1 (hEAAT1) residues highlighted in the simulations as part of the chloride permeable pathway. The anion current mediated by hEAAT1 when Cl$^-$ serves as the main permeant anion is barely detectable (*Torres-Salazar et al., 2015*), therefore more permeant anions such as NO$_3^-$ or SCN$^-$ are commonly used when studying the anion conductance in EAATs (*Machtens et al., 2011*; *Ryan et al., 2004*; *Watzke and Grewer, 2001*). Thus, we measured NO$_3^-$ and/or SCN$^-$ currents of these mutants expressed in *Xenopus* oocytes, explored their accessibility to thiol-specific reagents and tested the effect of the modification on the anion permeation and substrate transport. During the application of thiol-modifying reagents, we measured currents at a constant voltage of +60 mV at 1 s intervals to monitor any changes and to obtain a modification rate (e.g. *Figure 3C*). In addition, the anion current amplitude, the glutamate transport current and radiolabeled glutamate uptake was measured under the same conditions before and after the application to compare the effect of the modifications on anion currents and substrate transport under identical conditions (e.g. *Figure 3D and E*).

In oocytes expressing M89C (V51 in Glt$_{Ph}$), using a NO$_3^-$-based solution to monitor the effect on the anion conductance, application of 10 mM MTSES decreased the anion current amplitude at +60 mV to 61.5 ± 5% with a time constant of 5.5 ± 0.5 s (n = 6) (*Figure 3C and D*), an effect that was reversed by the application of 1 mM 1,4-Dithiothreitol (DTT) (*Figure 3D*). This reduction in current amplitude occurred at positive potentials, suggesting that only the anion conductance was affected and that substrate transport, which is predominant at negative potentials, remained intact (*Figure 3* and *Figure 3—figure supplement 1*).

In Cl$^-$-based solutions, subtraction of the current in the absence of glutamate from the current in its presence results in a current-voltage relationship that predominantly reflects the glutamate transport current, displaying strong rectification and negligible currents at positive potentials. To corroborate the effect of the modification on substrate translocation, we measured in the same cells, the current in the absence and the presence of saturating concentrations of glutamate (500 μM) using a Cl$^-$-based solution before and after application of the thiol-reagent. In contrast to the anion current, the transport current in cells expressing M89C remained intact after application of MTSES (*Figure 3—figure supplement 1*). Moreover, the uptake of radiolabeled glutamate was not affected by application of 10 mM MTSES to oocytes expressing M89C (*Figure 3E*, n > 15). These experiments demonstrate that while application of MTSES significantly decreased anion permeation, it did not affect glutamate translocation. To rule out the possibility that the electrostatic interactions of the negatively charged MTSES might affect anion permeation, we repeated the same experiment using the positively charged MTSET and the neutral MTSACE (*Reyes and Gadsby, 2006*). Both reagents produced a similar decrease in the anion current without altering substrate translocation (*Figure 3F–K*, n > 3), further confirming that M89C is accessible to thiol-specific reagents and, that the resulting modifications decrease anion permeation without affecting substrate translocation.

We obtained similar results with L296C (L212 in Glt$_{Ph}$), another residue highlighted in our simulations to be exposed upon opening of the anion pore in the $iChS$. Application of 10 mM MTSES to cells expressing L296C also reduced the NO$_3^-$-mediated current at +60 mV to 59 ± 8.9% ($n > 5$) with a time constant of 15.6 ± 0.5 s (*Figure 3L–M*), which was reversible by DTT (*Figure 3M*) and did not affect glutamate uptake (*Figure 3N*). MTSACE also reduced the anion current without altering substrate transport (*Figure 3O–Q*, $n > 3$).

A third residue that our simulations revealed as part of the anion permeation pathway was L88 (F50 in Glt$_{Ph}$) (*Figure 2*). Notably, our SASA calculations revealed that at least in hEAAT1, L88 was the residue that showed the most dramatic change in its SASA between the $iOFS$ or IFS and $iChS$ (*Figure 3B*). In the IFS and $iOFS$, the SASA for L88 was less than 10 Å$^2$, while in the $iChS$, it became 60 ± 20 Å$^2$ (comparable to that of M89 and L296). In contrast to oocytes expressing M89C or L296C, the oocytes expressing L88C did not show any significant changes in current amplitudes or substrate transport upon application of 10 mM MTSES or 1 mM MTSET (*Figure 4A–B,D–E and I*), suggesting that L88 was not accessible from the EC side of the membrane. To test for potential accessibility from the IC side, we applied the membrane permeable MTSEA (*Holmgren et al., 1996*) to oocytes expressing this mutant. Application of 2.5 mM MTSEA resulted in a slow increase in the anion current amplitude at +60 mV. About 10 s after the application, the current started increasing, to reach a maximum of 10-fold increase within ~50 s (*Figure 4C*, open circles, $n = 6$). No detectable changes were observed in oocytes-expressing Csls WT 50 s after an identical application of MTSEA (*Figure 4C*; gray triangles, $n = 3$). The full current-voltage relationships from −120 to 60 mV measured before and 3 min after the MTS-application confirmed the effects of the modifications observed at +60 mV (*Figure 4D–F*). To test whether L88C was not accessible to MTSES and MTSET, or if it was accessible but the modification caused no effect on the carrier function, we designed a new experiment in which both reagents were independently applied before the application of MTSEA. If the initially applied reagent could access L88C, the subsequent application of MTSEA would have no effect or at least a significantly lower effect than it had when applied without previous treatment. *Figure 4G and H* ($n > 3$) show that the effect of MTSEA remains unchanged when applied after a previous application of the other reagents, indicating that L88C was not accessible to MTSES or MTSET. The lack of reactivity of L88C to membrane impermeable reagents suggests that L88C is deep in the anion pore and probably accessible only from the IC side.

Interestingly, modification of L88C with MTSEA not only increased macroscopic anion currents but also significantly decreased substrate transport. Radiolabeled glutamate uptake was reduced more than 60% after incubation with 2.5 mM MTSEA (*Figure 4I*), the same conditions that resulted in a 10-fold increase of anion current. A modification that increases anion conductance and simultaneously decreases substrate transport is typical of residues involved in anion channel gating (*Borre et al., 2002*; *Hotzy et al., 2012*; *Seal et al., 2001*; *Shabaneh et al., 2014*; *Torres-Salazar et al., 2015*). Therefore, these results suggest that L88 may be part of the permeation pathway, but it is also involved in anion channel gating. Because MTSEA is positively charged, in an attempt to mimic the effect of the modification, we substituted L88 by arginine and measured macroscopic current amplitudes, radiolabeled glutamate uptake as well as relative permeability ratios. Similar to that observed after the modification with MTSEA, in oocytes-expressing L88R, we observed much larger macroscopic current amplitudes. The increase was more than 10-fold when normalized by surface expression (*Figure 4K*, $n > 10$). Moreover, L88R showed less than 90% of substrate transport activity when compared with WT (*Figure 4L*, $n > 20$).

Both the modification of L88C with MTSEA and the substitution L88R suggest that L88 plays a role in the coupling of anion channel gating and substrate transport as previously observed with other modifications/substitutions (*Borre et al., 2002*; *Hotzy et al., 2012*; *Seal et al., 2001*; *Shabaneh et al., 2014*; *Torres-Salazar et al., 2015*). Despite the evidence that L88 may be playing a role in anion channel gating, our simulations and SASA estimations also directly implicate L88 in direct contact with the permeant anions (*Figures 2C,D* and *3B*), an observation that is also consistent with a previous report (*Machtens et al., 2015*). To corroborate this, we examined the permeability ratios of the mutant L88R and compared with those of WT. As shown in *Figure 4M*, the permeability ratios to chloride in oocytes expressing the mutant L88R were significantly higher for all permeant anions when compared to the WT permeability ratios, demonstrating that L88 may also be playing a role in anion selectivity. These results are in perfect agreement with previous observations showing that a similar substitution in the homologous residue in Glt$_{Ph}$ (F50K), results in a

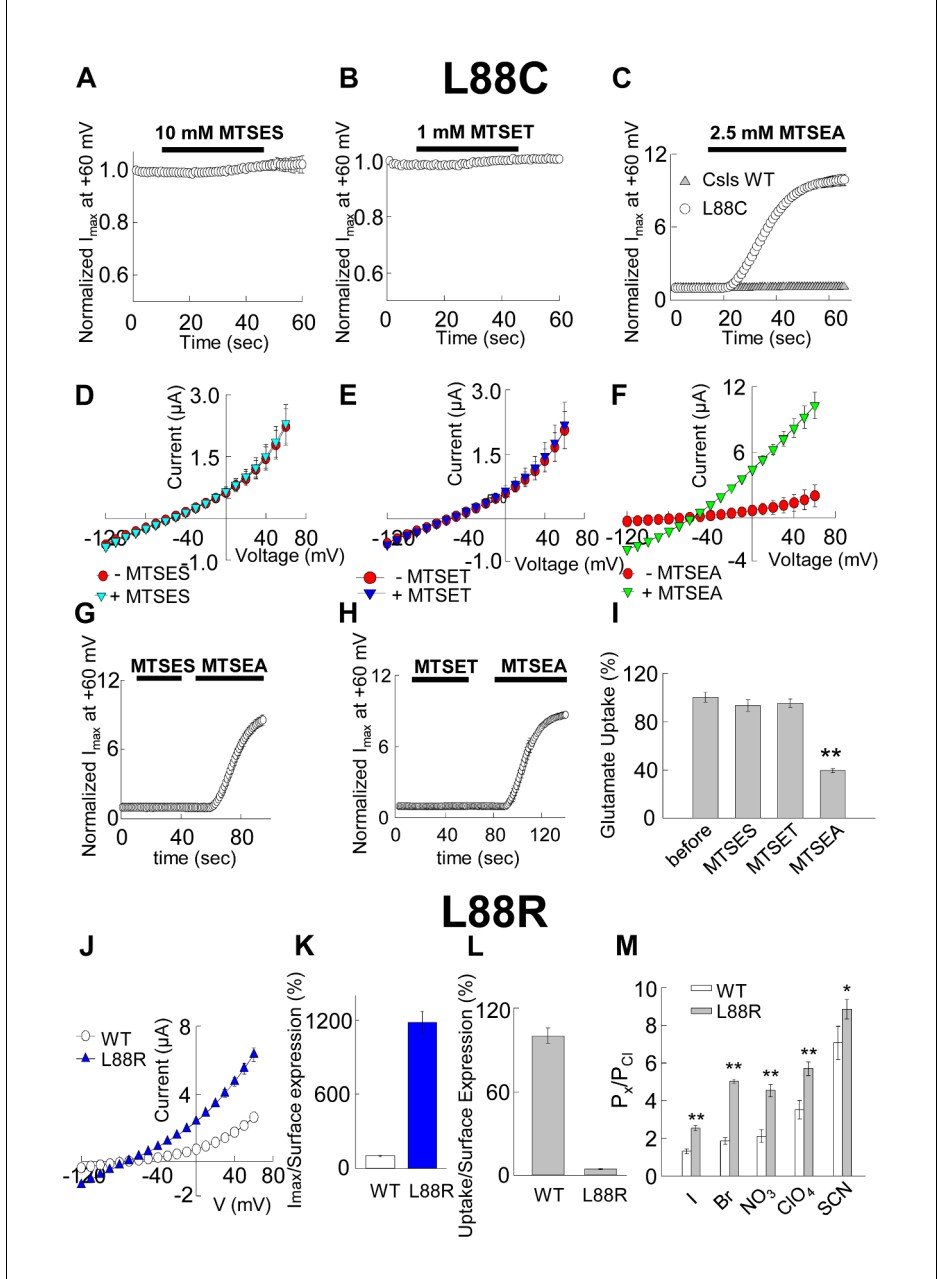

**Figure 4.** Accessibility and permeation properties for EAAT1 L88. (**A–C**) Averaged normalized current amplitude measured at constant voltage (+60 mV) in oocytes-expressing L88C. The voltage pulse was continuously applied at 1 s intervals to monitor the effect of the application of 10 mM MTSES (**A**, $n = 4$), 1 mM MTSET (**B**, $n = 4$) or 2.5 mM MTSEA (**C**, $n = 6$) on the anion conductance. The *black horizontal bar* represents the application duration of MTS-reagent. Panel **C** shows the averaged normalized current amplitude measured in oocytes expressing Csls WT (*grey triangles*, $n = 3$) in response to an identical application of MTSEA. (**D–F**) Averaged current-voltage relationships measured in the same cells showed in (**A–C**) before (*red circles*) and after (*green inverted triangles*) application of the different MTS-reagents. (**G–H**) Averaged normalized current amplitude measured at a constant voltage at +60 mV in oocytes expressing L88C to monitor the modification of MTSEA after a previous application of MTSES (**G**) or MTSET (**H**). (**I**) Radiolabeled glutamate uptake measured in oocytes expressing L88C before and after incubation with the different MTS-reagents for the same time and under the same ionic conditions used in **A–C**. At least 25 oocytes were included for each group. Background radiolabeled glutamate accumulation measured in water-injected oocytes was subtracted from each group. (**J**) Current-voltage relationship measured in oocytes expressing WT EAAT1 (*open circles, n = 10*) or L88R (*blue triangles, n = 12*). (**K**) The maximum current amplitude at +60 mV from panel **J** was normalized by surface expression to reflect the actual difference in current amplitude. (**L**) The

*Figure 4 continued on next page*

*Figure 4 continued*

same surface expression was used to normalize radiolabeled glutamate uptake measured in oocytes-expressing WT EAAT1 (*n* = 20) or L88R (*n* = 22). For the surface expression, equal groups of 10 oocytes were used and the experiment was repeated two times in two different batches of oocytes. (M) Relative permeability ratios measured in oocytes-expressing WT EAAT1 (*n* = 7) or L88R (*n* = 8).

significantly larger anion conductance (about fourfold) and that negatively charged substitutions at the same position in Glt$_{Ph}$ and EAAT2 affect the cation/anion permeability of the channel (*Machtens et al., 2015*).

## The anion-channeling pathway emerging in the *iChS* is energetically favored over other putative pathways

At present, there is still a debate about the actual Cl⁻ permeation pathway(s) in glutamate transporters (for a review, see [*Cater et al., 2016*; *Fahlke et al., 2016*]). We examined the probabilistic occurrence of alternative Cl⁻ pathways suggested in the literature by systematic metadynamics simulations (*Laio and Parrinello, 2002*). The idea is to 'direct' the anion along particular pathways and evaluate the free-energy profile (or the potential of mean force, PMF) along those pathways.

First, we examined the energetics of Cl⁻ permeation across the water channel observed in our *iChS* protomer (*Figure 5A*). The *cyan dots* therein refer to the instantaneous positions of the chloride ions from multiple runs as they are channeled through that pathway. *Video 2* displays the trajectory of a Cl⁻ ion observed in metadynamics simulations. We calculated the PMF profiles for the permeation of chloride or sodium ions across the water channel (see *Figure 5D*). The estimated energy barrier is ~4.0 kcal/mol, as Cl⁻ translocates through the hydrophobic constriction zone lined by F50, V51 and L212 (*Figure 5C* and *Video 2*). Notably, R276 near the IC entrance plays a significant role in selectively permeating Cl⁻ over Na⁺ (*Figure 5D*). In this region, Na⁺ ions encounter an energy barrier of 8.0 kcal/mol, indicating that the anion permeation path in the *iChS* is practically not accessible to Na⁺ ions. *Video 2* also shows that P356 interacts with the translocated Na⁺ during the transition to the anion-channeling state.

Second, we explored the putative chloride paths lined by S65 in Glt$_{Ph}$. S65 in Glt$_{Ph}$ (*Ryan and Mindell, 2007*) as well as its counterpart S103 in EAAT1 (*Ryan et al., 2004*) were reported to be important to Cl⁻ permeation. Mutation S65V in Glt$_{Ph}$ significantly reduced Cl⁻ flux, with no obvious effects on the substrate transport, but impaired the ability of Na⁺/aspartate activation of anion permeation (*Ryan and Mindell, 2007*). Our simulations initiated with a Cl⁻ ion originally placed near S65 in the MD-equilibrated *iOFS* showed that the Cl⁻ ion migrated into the bound Na2 site, via a trajectory (*Figure 5B* and *Video 3*) broadly consistent with the putative anion permeation pathway suggested by Verdon and Boudker (*Verdon and Boudker, 2012*). However, the path remained obstructed by the hydrophobic residues V58, I61, V355, and A353, resulting in an energy barrier over 20 kcal/mol (not shown). This result shows that this path is highly improbable to allow anion permeation, and S65 is not likely to direct the chloride ion to a permeable channel. The putative permeation pathway involving S65 remained closed at all times in our simulations. Moreover, in oocytes-expressing S103C, we did not observe any significant modification effect on the anion current upon application of MTSET, MTSES or MTSEA (*Figure 5—figure supplement 1*).

## The chloride channeling region is separated by a flexible wall from the substrate-binding site

We note that in our simulations, the initial formation of the water channel occurred in a substrate-loaded state, in which the substrate-binding site was minimally hydrated and practically occluded to both EC and IC regions. This differs significantly from the high level of hydration observed in the *iOFS* or IFS protomer, in which the substrate binding site is exposed to either the EC or the IC solution. The opening of the channel (*Figure 2*) is enabled by an 'elevator-like' displacement of the substrate-binding core (*Video 1*) and further enhanced by the applied voltage. The anion channel is positioned precisely at the interface between this 'moving' transport core and the trimerization domain. Several residues in HP1, HP2 and TM8 act as a flexible wall (*Figure 6*) between the substrate-binding site and anion-channeling path. The downward movement of the substrate-loaded

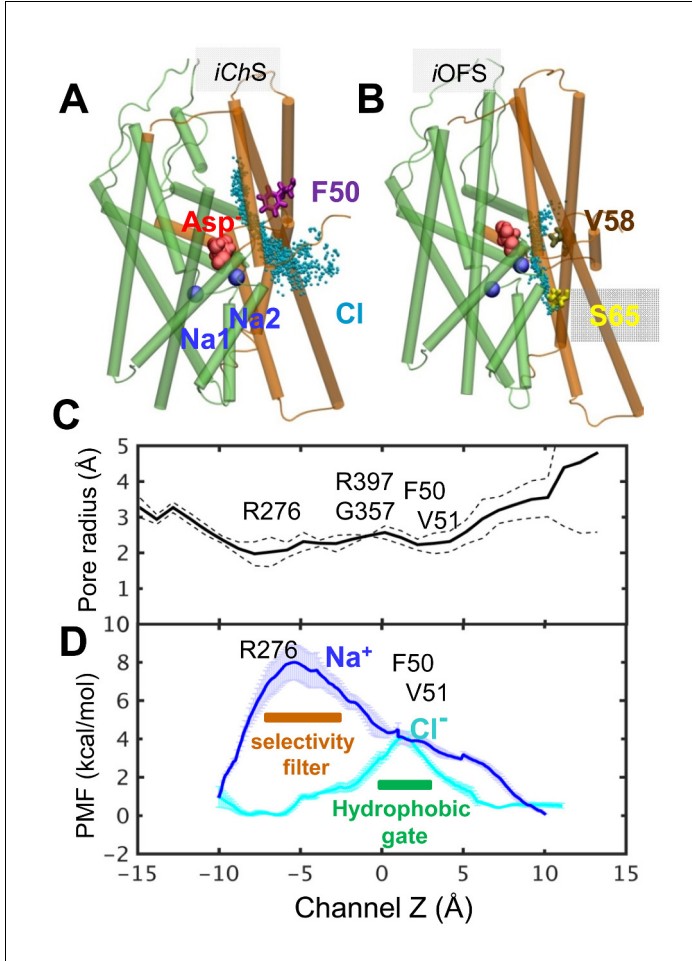

**Figure 5.** Chloride channeling pathway and its energetics. (**A**) Chloride permeation trajectories (*cyan points*) in the *iCh*S sampled by metadynamics performed for Glt$_{Ph}$. The energy barrier for chloride permeation is less than 5 kcal/mol; (**B**) An alternative pathway lined by V58, I61 and S65, proposed earlier to serve as anion-channeling path, when the subunit in the *iOFS*. The energy barrier is larger than 20 kcal/mol through this path; (**C**) *iCh*S channel pore radius along the z-axis, averaged out over multiple snapshots with dashed line showing the variations; and (**D**) potential of the mean force (PMF) for chloride (*cyan curve*) and sodium (*blue curve*) transport through the hydrated channel of the subunit in the *iCh*S, calculated by adaptive biasing force (ABF) method (**Chipot and Hénin, 2005**). The channel selectivity filter resides near the IC entrance (i.e. R276), and the potential channel gate is near the EC entrance formed by hydrophobic residues F50, V51 and L212 (see **Figure 2**).

The following figure supplement is available for figure 5:

**Figure supplement 1.** Application of 10 mM MTSES or 2.5 mM MTSEA did not modify anion current amplitudes in EAAT1 S103C (homologous to S65 in Glt$_{Ph}$).

core and especially these residues that form the wall create a volume expansion in the adjoining interfacial region, leading to the emergence of a continuous channel (**Figure 6**, and **Video 2**).

Toward a quantitative characterization of the difference between the *iOFS* structure resolved by X-ray crystallography (**Verdon and Boudker, 2012**) and the anion-channeling intermediate conformer *iCh*S observed in our simulations, we calculated the z-distance between the helical hairpins (HP1 and HP2) and S65. As shown in **Figure 7A–C**, the *iCh*S protomer exhibits intermediate, but clearly distinctive properties, compared to the *iOFS* and IFS. **Figure 3A–B** also shows the higher SASAs of channel-lining residues in the *iCh*S state, compared to their values in the *iOFS* and IFS. Furthermore, while the RMSD between *iOFS* and IFS is 7.3 Å, the *iCh*S exhibited RMSDs of 4.5 ± 0.5 Å and 3.2 ± 0.5 Å from the *iOFS* and IFS, respectively.

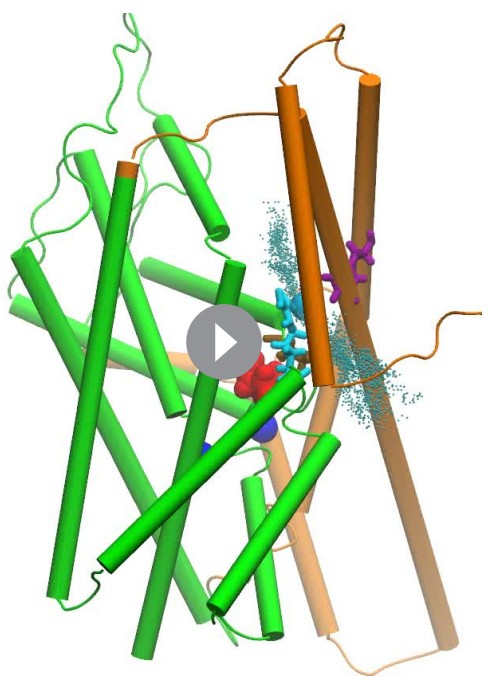

**Video 2.** Chloride channeling pathway. Metadynamics simulation of the passage of a chloride ion (*cyan*) through the Glt$_{Ph}$ subunit in the newly identified anion-conducting intermediate, *iChS*. In this case, we observe the translocation from the extracellular (EC) to the intracellular (IC) region. *Cyan dots* represent the trajectory of the chloride ions, sampled during the simulations. In the movie, F50, R276 and P356 are shown *purple*, *cyan* and *tan* stick representations. The bound aspartate and two sodium ions are shown in *red space-filling* and *blue spheres*, respectively. Note that, the chloride pathway is distinct from substrate/sodium-binding sites, and it is at the interface between the trimerization (*brown*) and transport (*green*) domains of the protomer.

**Video 3.** Putative pathway near S65 does not permit for chloride channeling. Sampling the vicinity of S65-associated cavity by chloride ion (*cyan*) near the *iOFS* using metadynamics simulation. *Cyan dots* represent the trajectory of the chloride ion, sampled during the simulations. The pore lining residues S65, I61 and V58 are shown *yellow*, *green* and *tan* stick representations. The bound aspartate and two sodium ions are shown in *red* and *blue space-filling*. The estimated energy barrier was over 20 kcal/mol through this path, and therefore we concluded that this is not a probable anion-channeling path.

## Modification of L88C and M89C is state-dependent

In our simulations, we only observed a continuous aqueous pathway in the *iChS*. This continuity was limited by constricted zones observed in the EC and IC regions in both the *iOFS* and IFS (*Figure 6A and C*). Because these constrictions may affect the accessibility of residues within the pore that are passed these points, we hypothesized that the accessibility of at least some of the pore forming residues would be affected when favoring these conformations where we observed an interrupted continuous pathway. To evaluate our hypothesis, we repeated the modification experiments in conditions that would favor the stabilization of the carrier in either an inward (in high extracellular K$^+$ [*Shlaifer and Kanner, 2007*]) or an outward (in the presence of TBOA [*Boudker et al., 2007*]) conformation. In oocytes-expressing M89C, we applied 10 mM MTSES in a KNO$_3$-based solution, which is expected to favor inward-facing conformations and consequently a constriction in the EC region of the pore (*Figure 6C*). When we measured the current amplitudes in NaNO$_3$ + glutamate before (*white bars*) and after (*black bars*) the application of MTSES in high K$^+$ conditions (*Figure 8A*, *center bars*, n = 5), we did not observe the 40% reduction of the current amplitudes obtained when the application was done in conditions that favored channel opening (*Figures 3C* and *8A* right bars, n = 5). We performed a similar experiment in oocytes-expressing L88C but using MTSEA, which we previously demonstrated that it modified L88C from the IC part (*Figure 4*). Although application of

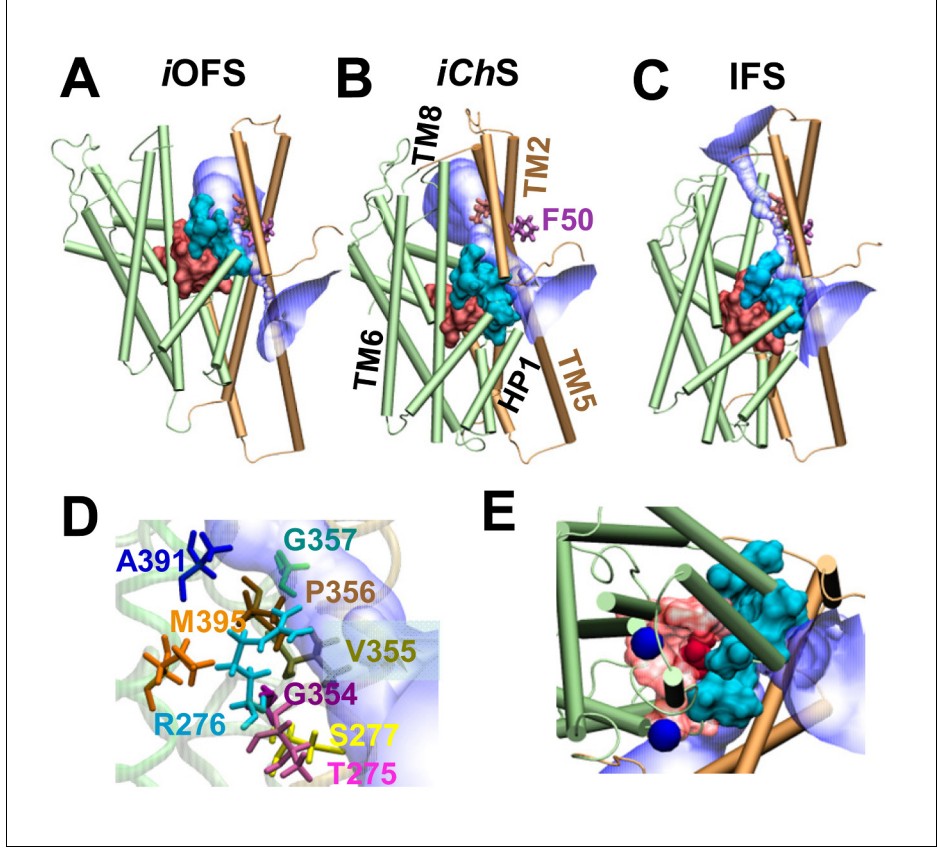

**Figure 6.** A moving flexible wall separates the substrate-binding and anion permeation regions. The wall (shown in *cyan surface* representation) is composed of residues T275-R276-S277 in the HP1 loop, G354-V355-P356-G357 in the HP2 loop, and A391, D394 and M395 in the TM8. The substrate-binding cavity (composed of residues located within 4 Å from the bound aspartate) is shown in *red surface* representation. The anion permeation region is shown in *semi-transparent blue/white*. The structures are shown for (**A**) *i*OFS; (**B**) *i*ChS; and (**C**) IFS in $Glt_{Ph.}$ TM helices belonging to the transport domain are colored *green*; those in the trimerization domain are *light brown*. Comparison of panes **A–C** shows that the trimerization domain remains approximately fixed in space, while the transport domain moves downward (elevator-like) as the subunit transitions from *i*OFS to IFS. Panel (**D**) shows the wall-composing residues in *i*ChS; and (**E**) displays the IC view of the wall-composing residues (*cyan surface*), bound sodium ions (*blue spheres*) and aspartate (*red VDW* representation). Note that only in the *i*ChS conformer is the channel wide enough to allow for the permeation of anions. In both panels **A** and **C**, there are constriction zones near F50, which block the path for anion permeation.

MTSEA in a KNO$_3$-based solution did not prevent the modification (*Figure 8B*, *central bars*, n = 4), its effect was much smaller than the observed when MTSEA was applied in a NaNO$_3$-based solution (*Figures 4* and *8B*, right bars, n = 5). If we consider that MTSEA access L88C from the cytoplasm and that the constriction observed at the IC part of the channel in the IFS is not as small as the one observed at the EC region (*Figure 6C*), a partial modification of L88C is not surprising. On the other hand, based on the tight constriction zone observed at the IC region in the *i*OFS (*Figure 6A*), one would expect a much reduced or no accessibility of MTSEA to L88C under conditions that favor this conformation. To test that, we then applied MTSEA in the presence of TBOA, which lock the carrier in an outward or intermediate outward conformation (*Boudker et al., 2007*). In the presence of 200 µM TBOA, no effect of MTSEA was observed (*Figure 8C–D*, n = 5). In the same cells, following the application in TBOA, we applied MTSEA in the same conditions used before (*Figure 4*) and the effect was very similar, increasing the current about 10-fold (*Figure 8C–D*). This result demonstrates very little or no modification at all in the presence of TBOA, supporting the hypothesis that when the carrier is stabilized in an outward-facing conformation the accessibility of L88C from the

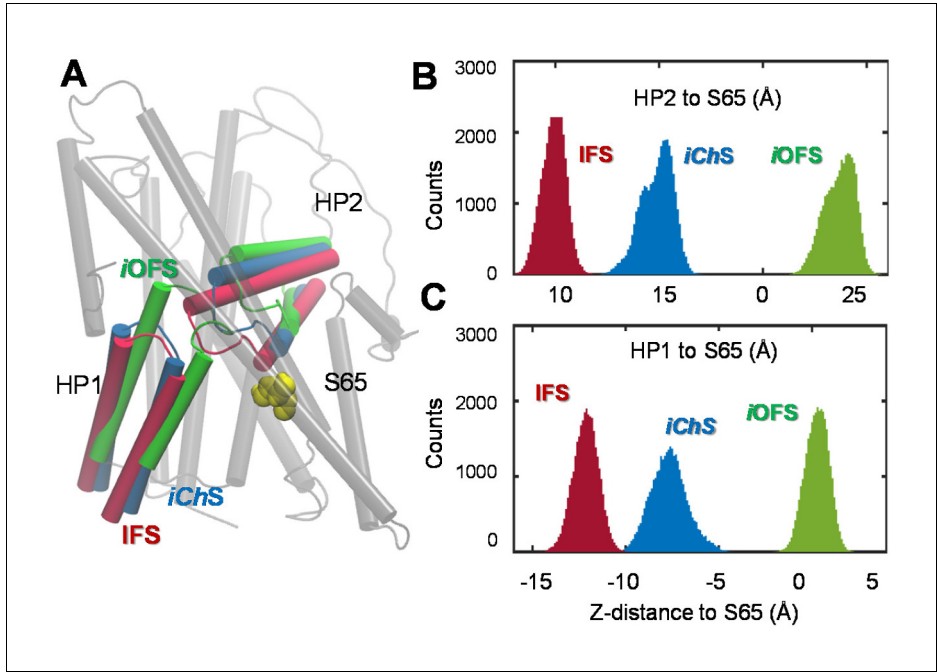

**Figure 7.** Distinct characteristics of the channeling intermediate *iCh*S stabilized during the transition from *iOFS* to IFS in Glt$_{Ph}$. (**A**) Structural superimposition of *iCh*S (*blue*) with the crystal structures resolved for the *iOFS* (*green;*) and IFS (*red;*). Histograms of the *z*-distance between (**B**) the external gate HP2 and TM2 (S65); and (**C**) the inner gate HP1 and TM2 (S65), based on 320,000 MD snapshots. The distances are based on the *z*-coordinates of the mass centers of HP2 (V335-V370), HP1 (P258- K290) and S65.

cytoplasm is limited or abolished, consistent with the constriction we observed in the IC region in a similar conformation (*Figure 6A*). To ensure that TBOA was working, we measured the current amplitudes in the absence and the presence of 200 μM TBOA before and after each of the applications; in all cases, TBOA blocked the anion current near background levels (*Figure 8E*).

## Discussion

Increasing evidence for the involvement of EAAT-associated anion channels in the regulation of glutamatergic synaptic transmission (*Melzer et al., 2005*; *Veruki et al., 2006*; *Wersinger et al., 2006*) and their direct association with neurological disorders (*Parinejad et al., 2016*; *Winter et al., 2012*), underscore the importance of establishing the molecular determinants of EAATs anion channel function. The present study highlights the high propensity of glutamate transporters to form an anion-selective channel upon reconfiguration of the substrate-loaded transporter. The anion selective pore, consistently observed in simulations of the archaeal aspartate transporter, Glt$_{Ph}$, and confirmed for the homology-modeled human EAAT1 (*Figures 1* and *2*), became accessible when the transporter visited an intermediate state, *iCh*S, during its transition from *iOFS* to IFS. This intermediate state is distinguished by the repacking of a hydrophobic cluster (*Figure 2*) that promotes the opening of a continuous cavity between the transport and the trimerization domains, which, in turn, facilitates anion permeation, in agreement with previous reports (*Cater et al., 2014*; *Machtens et al., 2015*). The channeling intermediate *iCh*S is distinct from the X-ray resolved OFS, *iOFS* and IFS (*Figures 3A–B* and *7*) in Glt$_{Ph}$. The versatility of the transport domain to sample yet another functional conformation relative to the relatively rigid trimerization domain is consistent with the conformational heterogeneity of Glt$_{Ph}$ transport noted by Slotboom and coworkers in their EPR study (*Hänelt et al., 2013*).

A combination of electrophysiological recordings with cysteine accessibility assays in *Xenopus* oocytes (*Figures 3C–Q* and *4*) confirmed the critical role of the conformational flexibility of the pore-lining residues near the EC entrance of the pore, as observed in our simulations. In oocytes-

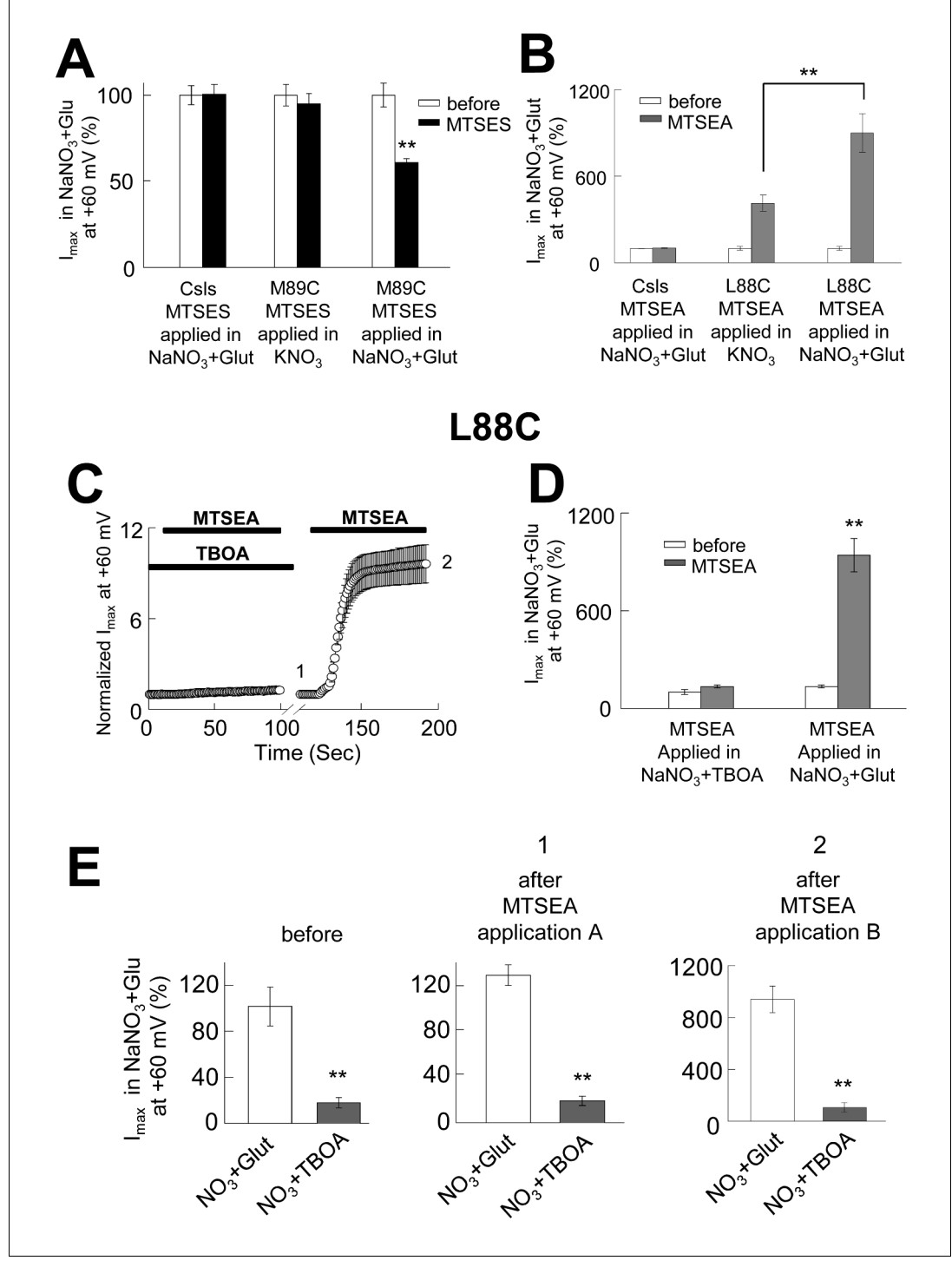

**Figure 8.** The opening of the anion permeation path occurs only in the *iChS*, and the accessibility of pore-forming residues is state-dependent. (**A**) Bar graph representation of averaged current amplitudes at +60 mV measured in oocytes expressing either CslsWT or M89C before (*white bars*) and after (*black bars*) the application of 10 mM MTSES. MTSES was applied in a buffer containing 96 mM $KNO_3$ and no sodium (center bars, $n = 5$) or in a buffer containing 96 mM $NaNO_3$ +500 μM glutamate (right bars, $n = 4$). (**B**) Bar graph representation of the averaged current amplitudes measured in oocytes expressing either CslsWT or L88C before (*white bars*) and after (*grey bars*) the application of 2.5 mM MTSEA. The application was done in a buffer containing $KNO_3$-based solution ($n = 5$) or in a $NaNO_3$-based solution +500 μM glutamate. (**C**) Averaged normalized current amplitudes measured in oocytes expressing L88C at +60 mV on intervals of 1 s before and during the application of 2.5 mM MTSEA. A first

*Figure 8 continued on next page*

*Figure 8 continued*

application was done in the presence of 200 µM TBOA, followed by a second application in NaNO$_3$+glutamate ($n$ = 5). The black bars represent the time of each application. (**D**) Bar graph representation of averaged current amplitudes at +60 mV measured before the application (*white bars*) and after the first (1) and the second (2) application in C) (*grey bars*). (**E**) Averaged current amplitudes at +60 mV in the absence and the presence of 200 µM TBOA before and after each of the applications in (**C**).

expressing mutants of introduced cysteines in predicted pore-forming residues, we observed a reduction of the macroscopic current amplitude by about 50% after application of different MTS-reagents. Moreover, our results showed that most modifications that significantly affected anion permeation did not affect transport currents or radiolabeled glutamate uptake, indicating that substrate transport is not altered by these modifications (*Figure 2*). The two residues that showed a reduction in current amplitudes upon modification with MTS-reagents without affecting substrate translocation (M89 and L296), were the ones that displayed the highest solvent accessibilities as reflected by their high SASA values in the iChS. They also have significantly lower SASA values in the IFS or the iOFS (*Figure 3B*), confirming that their aqueous exposure to the solvent reaches a maximum when the channel adopts the open state. It is worth noting that M89C and L296C are homologous of V51 and L212 in Glt$_{Ph}$, which in the report by Machtens et al. (2015) showed the highest interaction with Iodide and the largest chloride occupancy in their simulated open channel conformation. We did not observe changes in current amplitude with any of the MTS-reagents applied to other cysteines substituted residues predicted to be pore-lining in MD simulations (I293C, M286C, R477C, S363C or M279C, data not shown). These residues are either deeper into the pore (e.g. I293 and M286) and inaccessible to MTS-modification, or located on the wide open surface of the channel (e.g. M279 and R280) such that the modification with bulky reagents would not affect anion permeation. R276 in HP1 of Glt$_{Ph}$ (homologous S363 in EAAT1) and a highly conserved basic residue in TM8 of EAATs (R477 in EAAT1) have been suggested to line the anion pore and serve as selectivity filters for the channel (*Machtens et al., 2015*). Indeed, in the Glt$_{Ph}$ iChS, R276 is a critical residue that prevents the permeation of Na$^+$ through the anion channel (*Figure 5D*). Although EAAT1 R477 is highly exposed to water in the *iCh*S (SASA value of 110 ± 40 Å$^2$) and IFS (SASA value of 75 ± 25 Å$^2$), its substitution by cysteine in the R477C mutant (data not shown) decreased the SASA to 30 ± 10 Å$^2$, which may explain the limited accessibility to modifying reagents (data not shown). In fact, only the residues that show a high SASA in the *iCh*S and low SASA in IFS and/or *i*OFS could be modified by MTS-compounds. Another substituted cysteine mutant, L88C, could only be accessed from the IC side of the membrane. Modification of L88C affected both anion currents and substrate transport, suggesting that L88 resides deep within the pore and may participate in channel gating and line the permeation pathway. Interestingly, a very similar phenomenon is observed in voltage-gated chloride channels (ClCs). In ClC-0, specific residues known to interact with the permeant anion contribute to both anion conduction and channel gating (*Pusch et al., 1995*). As shown for ClCs, the gating of EAAT-associated anion channels is also modulated by permeant anions (*Kovermann et al., 2010*; *Melzer et al., 2003*; *Torres-Salazar and Fahlke, 2006*). We suggest L88, in addition to coordinating anion permeation through the pore, may also regulate a shift in the equilibrium between channel gating and substrate translocation cycle.

During transition from *i*OFS, a continuous water channel was initially formed when the RMSD of the conformer was around 3.5 ~ 4.0 Å away from the *i*OFS and was dehydrated once the protomer approached the IFS (e.g. RMSD >5.5 Å). The observation of an intermittent water channel during the global transition of the Glt$_{Ph}$ complies with the notion that water conducting-states may generally arise as short-lived intermediates as the protein undergoes a transition from one global state to another (*Li et al., 2013*). Recently, Kanner and coworkers reported that cysteine cross-linking of the W441C/K269C double mutant of the neuronal glutamate transporter EAAC1 allowed for anion conductance, but not substrate transport (*Shabaneh et al., 2014*). Similarly, constraining the homologous Glt$_{Ph}$V216-A391 distance in silico restricted the lateral motions of the transporter domain but allowed for sufficient conformational flexibility to form an anion-conducting pore (*Machtens et al., 2015*). Notably in our simulations, the initial formation of the water channel occurred as the C$^\alpha$ distance of V216-A391 was within 9.5–11.5 Å, therefore the cysteines in the double mutant V216C/

A391C are likely to form cross-links. We also recently identified a point mutation in the IC end of TM7 (R388D/E) that drives the carrier into a constitutive open channel state while virtually eliminating substrate transport (*Torres-Salazar et al., 2015*).

In addition, our data identified residues in HP1 and HP2 loops and TM8, e.g. R276, P356, G357, A360 and A391 that act as a 'moving wall' (*Figure 6*) between substrate-transport cavity and the anion permeation channel. This underscores the idea that the channel and substrate translocation pathways are mutually exclusive. Another interesting and novel observation is the constriction zones that we observed of the permeation pathway in the *i*OFS and the IFS, which explains why the continuous permeation pathway is only observed in the i*ChS* (*Figure 6B*)(*Machtens et al., 2015*). This was confirmed by the state-dependent modification that we observed with two of the MTS-reactive residues (L88C and M89C). Perfusion of the cells with a $K^+$-based external solution shifts the equilibrium of the carriers toward the IFS. When MTS-reagents were applied under this condition, M89C became inaccessible (*Figure 8A*), and the IC accessibility of L88C was only partially reduced (*Figure 8B*). As predicted from the pathway associated with the IFS, there is a tighter constriction of the cavity on the EC side and a more open configuration on the IC (*Figure 6C*). However, when we repeated the experiment in the presence of TBOA, which lock the protein in an OFS or iOFS (*Boudker et al., 2007*), the accessibility of M89C was not significantly altered (data not shown), and L88C became inaccessible (*Figure 8C–D*), findings consistent with the very tight constriction of the pathway on the IC side in the *i*OFS (*Figure 6A*). These data support the state-dependent changes in the shape of the pore and illustrate why channel opening in Glt$_{Ph}$ and EAAT1 occurs only in conformational states between iOFS and IFS (*Cater et al., 2014*; *Machtens et al., 2015*; *Shabaneh et al., 2014*; *Torres-Salazar et al., 2015*).

Our study also provides an independent perspective, based on computational and experimental methods different from earlier studies, on a number of divergent conclusions regarding the specific residues lining the conduction pathway (*Cater et al., 2014*; *Machtens et al., 2015*). The pore lining residues in our in silico deduced chloride channel in Glt$_{Ph}$ and the modeled hEAAT1, generally agree with those reported by *Machtens et al. (2015)*. One important discrepancy between previous studies was on the relevance of S65 (S103 in hEAAT1) to the formation of the permeation pathway. This residue was first observed to alter anion permeability ratios when mutated to valine in hEAAT1 (*Ryan et al., 2004*) and the same substitution was later confirmed to alter chloride flux in Glt$_{Ph}$ (*Ryan and Mindell, 2007*). These observations led the authors to propose S65 (S103) as a critical element of the chloride permeation pathway and the selectivity filter (*Cater et al., 2014*; *Ryan and Mindell, 2007*; *Ryan et al., 2004*), which was later supported by an aqueous cavity observed in a resolved crystal structure in the iOFS (*Verdon and Boudker, 2012*). Although S65 (S103) was not part of the continuous water pathway obtained in our simulations, we examined its role in greater detail by performing a series of simulations in the presence of a Cl$^-$ ion near S65, and additional experiments. In the simulations, although the Cl$^-$ ion initially moved into the crevice proposed as a potential pathway by Verdon and Boudker, this route remained obstructed with an energy barrier of more than 20 kcal/mol by hydrophobic contacts near I61 and V58. We further assessed the modification of the analogous residue in EAAT1, S103C, with different MTS-reagents (MTSES, MTSET or MTSEA), and did not observe an effect on current amplitudes with any of these reagents (*Figure 5—figure supplement 1*). Taken together with the previous reports, our results suggest that S65 (S103) may play a significant role in anion permeation by contributing to the structural elements involved in channel opening and closing but not as part of the permeation pathway.

Taken together, these data build on an emerging mechanism of anion channel pore opening by demonstrating that the open channel state proceeds through the cooperative dislocation of a flexible-wall during the elevator-like movement of the transport core. Our findings reinforce the idea that substrate transport and anion permeation proceed through two mutually exclusive pathways that are facilitated by conformational changes involving this flexible wall domain. The same mechanism is robustly shared by both the archaeal and mammalian transporters Glt$_{Ph}$ and EAAT1, and confirmed by both experiments and simulations. These efforts and those of others have begun to illuminate the anion permeation pathway and gating mechanism, and also provide a basis for exploring how EAAT-associated anion channels regulate synaptic function and contribute to neurological and neuropsychiatric conditions.

## Materials and methods

### Experimental

#### Heterologous expression of EAAT1

Point mutations were introduced into the hEAAT1 WT cysteineless cDNAs (Csls WT) using the Quik-Change mutagenesis kit (Stratagene, La Jolla, CA). WT and mutant hEAAT1-capped cRNA were synthesized from smal-linearized pOTV through use of MESSAGE machine kits (Ambion, Austin, TX). Subsequently, cRNA was resuspended in 10 µl of water then adjusted to a concentration of ~500 µg/ml and stored in 2 µl aliquots of at –80°C until use. Of ~500 µg/ml cRNA, 50 nl were injected into oocytes using a nanoliter 2000 injector (World Precision Instruments, Sarasota, F), and oocytes were kept at 18°C in ND-96 solution supplemented with 2.5 mM sodium pyruvate and 100 µg/ml gentamycin sulfate prior to recordings. Electrophysiological recordings and uptake assays were done 2 to 3 days after injection.

#### Electrophysiology

EAAT-associated currents were recorded in oocytes by two-electrode voltage clamp using a Dagan CA1B (Dagan Corporation, Minneapolis, MN) amplifier. $Cl^-$, $NO_3^-$ or $SCN^-$ were used as a main permeant anion. Recording solutions contained (in mM): 96 Na ($Cl^-$, $NO_3^-$ or $SCN^-$), 4 KCl, 0.3 $CaCl_2$, 1 $MgCl_2$, 5 HEPES, pH 7.4 in the absence or in the presence of 500 µM external glutamate. Oocytes were held at −60 mV, and currents were elicited by 200 ms voltage steps between –100 mV and +60 mV, filtered at 2 kHz (−3 dB), and digitalized with a sampling rate of 10 kHz using a Digidata AD/DA converter (Axon Instruments). The current-voltage relationship curves were plotted without using any current subtraction procedure. For cysteine modification experiments, oocytes-expressing Csls WT and mutant transporters were perfused or incubated for 3 min in ND96 buffer containing different sulfhydryl-modifying reagents. The concentrations used for the different reagents were: 1 mM for MTSET, 10 mM for MTSES, 1 mM for MTSACE and 2.5 to 5 mM for MTSEA. All reagents were diluted in water to 1M divided in aliquots and frozen for up to 6 months. All current recordings were additionally measured before and after the application of the reagents. In a set of experiments, 1 mM DTT was perfused after the modification to reduce the disulfide bridge.

#### Uptake assay

$H^3$-glutamate uptake assays were performed on *Xenopus* oocytes. For the cysteine modification experiments, oocytes were incubated for 3 min in ND96 buffer containing the different sulfhydryl-modifying reagents (with the same final concentration used in the electrophysiological recordings) and then washed with ND96 before proceeding with the experiment. The uptake buffer contained (in mM): 96 NaCl, 4 KCl, 0.3 $CaCl_2$, 1 $MgCl_2$, 5 HEPES, pH 7.4. Right before the experiment, labeled (500 nM) and unlabeled (500 µM) glutamate was added to reach saturating glutamate concentrations. Oocytes were incubated in radioactive buffer for 10 min, then transferred three times into ice cold ND96 containing (in mM: 96 NaCl, 4 KCl, 0.3 $CaCl_2$, 1 $MgCl_2$, 5 HEPES, pH 7.4) to stop transport activity. Immediately after, oocytes were individually transferred into scintillation counting vials containing 0.4 ml of 1% SDS. After gently shaking the vials for 1 hr, scintillation solution was added, and the samples were counted.

#### Data analysis

Two electrode voltage clamp data were analyzed using pClamp9 (Axon Instruments, Union City, CA) and the results from the electrophysiological recordings and uptake assays were analyzed using SigmaPlot (Jandel Scientific, San Rafael, CA). For statistical evaluation the Student's t-Test was used.

### Computational

#### Modeling of $Glt_{Ph}$

The initial simulation system of $Glt_{Ph}$ was constructed based on an asymmetric intermediate (PDB: 3V8G) (*Verdon and Boudker, 2012*) using VMD (*Humphrey et al., 1996*). The missing loops in the crystal structure were re-built using MODELLER (*Sali and Blundell, 1993*). All mutated residues in the crystal structure were reverted to their wild-type ones. Protonation states of titratable residues

were assigned based on pKa calculations using PROPKA (*Li et al., 2005*). Accordingly, D394 and D405 were protonated. Then the TM portion of the Glt$_{Ph}$ trimer was inserted into the center of a pre-equilibrated POPC membrane. Fully equilibrated TIP3 waters and 0.1 M NaCl were added to neutralize the system forming an all-atom simulation system of 135 Å × 135 Å × 95 Å. There were one Glt$_{Ph}$ trimer, 295 POPC, and about 27,766 water molecules for a total of over 141,800 atoms. The structure of the symmetric Glt$_{Ph}$ trimer in the IF state (PDB: 3KBC) (*Reyes et al., 2009*) embedded in the lipid bilayer was constructed following the similar procedure. *Figure 1A* illustrates the initial MD system set up. All MD simulations were performed using NAMD (*Phillips et al., 2005*) (version 2.9) and CHARMM36 force field with CMAP corrections (*Klauda et al., 2010*; *Mackerell et al., 2004*). The simulation protocol followed our previous approach (*Cheng and Bahar, 2013*, *2014*). MD runs are detailed in *Supplementary file 1*.

## Sampling of the conformational space near the *i*OFS of Glt$_{Ph}$

The initial MD system of *i*OFS (*Figure 1A*) was first energy-minimized for 50,000 step, followed by 0.5 ns constant volume and temperature (T = 310K) (NVT) simulations and a subsequent 4 ns Nosé-Hoover constant pressure and temperature (1 bar, 310 K) (NPT) simulation, during which the protein was fixed and constraints on the POPC head groups were gradually released. Subsequently, the constraints on the protein backbone were reduced from 10 kcal/mol to none within 3 ns. Finally, the unconstrained protein was subjected to NPT simulations. A total of 6 × 100 ns simulations were performed, among which two were in the absence of voltage and the other four in the presence of ±0.2 kcal/(mol Å e) (i.e. ±300 mV across the membrane) electric field. RMSD in C$^{\alpha}$-atom coordinates from their crystallized OFS positions reached a plateau of 3.0 ± 0.3 Å after 10 ns in all runs.

Six independent runs under three different conditions (in the absence and presence of externally applied electric fields of different strengths) invariably showed that putative pore suggested by Verdon and Boudker (*Verdon and Boudker, 2012*) remained 'closed' (with a radius of ~1 Å) near its constriction zone lined by M286, P304, S65 and Y195.

## Sampling structural transitions from *i*OFS to IFS and generation of channeling intermediate

We generated transport intermediates for the Glt$_{Ph}$ using the protocols adopted for characterizing leucine transporter intermediates along its transport cycle (*Cheng and Bahar, 2014*; *Shaikh and Tajkhorshid, 2010*). In particular, Glt$_{Ph}$ intermediates along the *i*OFS-to-IFS transition were derived using targeted MD (tMD) (*Schlitter et al., 1994*), starting from well-equilibrated asymmetric Glt$_{Ph}$ constructs. A steering force of $F_{tMD} = \frac{1}{2} (k/N) [(RMSD(t) - RMSD^*(t)]$ was applied to all atoms in the *i*OFS subunit (subunit C), using $k = 200$ kcal/(mol.Å$^2$); $N$ is the number of targeted atoms (backbone), RMSD(t) is the instantaneous departure from the target IFS, and RMSD*(t) is the target based on a linear decay from RMSD(0) to zero. Six runs were performed to sample transition of subunit C from *i*OFS to IFS. Along the *i*OFS-to-IFS transition, continuous water occupancy inside the subunit C was intermittently observed in all six runs. One channeling Glt$_{Ph}$ intermediate (*i*ChS; see *Figure 1B*) from tMD simulation was selected and further subjected to two 100 ns cMD simulations in the absence of voltage gradient. Moreover, four 100 ns cMD simulations were carried out, in which external electrostatic fields of ±0.1 kcal/(mol.Å.e) (±150 mV across the membrane) and ±0.3 kcal/(mol.Å.e) (±450 mV across the membrane) were applied, respectively. The water channel remained open in three different runs, in which the channel lining residues were broadly consistent despite some variations in the side chain orientations.

## Sampling chloride permeation pathways in Glt$_{Ph}$

Chloride permeation pathway and associated potential of mean force (PMF) were assessed using metadynamics (*Laio and Parrinello, 2002*) and adaptive biasing force (ABF) (*Chipot and Hénin, 2005*) methods implemented in NAMD. We used both metadynamics and ABF to estimate the free-energy profiles along the putative pathways, as a way of consolidating the results. These two methods complement each other. Metadynamics has the advantage of identifying curved permation pathways and estimating the free-energy profiles with less computational effort, albeit at lower accuracy compared to other methods (*Zhang and Voth, 2011*), like ABF. We used metadynamics for a first, quick assessment of putative chloride channeling pathways. Then the one with the lowest energy

barrier was reevaluated using ABF to achieve higher accuracy. Briefly, metadynamics simulations were carried out in six successive windows of width 4–6 Å along the channel ($z$-) axis (perpendicular to the membrane plane). Two consecutive 2 ns metadynamics runs were performed for each window. Using metadynamics, we sampled two Cl$^-$ pathways: (1) the equilibrated channeling intermediate (Meta_$iChS$ in *Supplementary file 1*); and (2) a path near S65 (Meta_S65). Following our previous approach (*Jun et al., 2016*), we calculated the PMF for Cl$^-$ or Na$^+$ permeation through the water channel using the ABF method. Briefly, PMF calculation started from the bulk solutions along the channel $z$ axis (perpendicular to the membrane lipids), which was subdivided into 4 ~ 5 different windows. The width of each ABF window was 5 Å and five to ten consecutive 1-ns ABF calculations were performed for each window until the variation of the PMF at any point along the z-axis was less than 2 kJ/mol within two consecutive runs. Overall, the calculated PMFs converged, as indicated by the <2 kJ/mol variation in PMF along the z-axis between two consecutive runs for permeant ions.

## Molecular dynamics (MD) study of the third sodium (Na3) binding

Experimental studies have found the co-transport of three sodium ions with one glutamate molecule (*Levy et al., 1998*; *Zerangue and Kavanaugh, 1996*). In the originally resolved crystal structures for Glt$_{Ph}$, there were two cation-binding sites (referred to as Na1 and Na2). The third sodium ion (Na3) was proposed to bind a site coordinated by T92, N310 and D312 in Glt$_{Ph}$ (*Bastug et al., 2012*; *Tao and Grewer, 2007*), which is now supported by the recent structure resolved for Glt$_{TK}$ (*Guskov et al., 2016*). Based on Glt$_{Ph}$ $iOFS$ conformer, we generated a simulation system in which the third sodium ion was docked onto this Na3-binding site (*Guskov et al., 2016*). Simulations in the presence of three sodium ions exhibited similar features for the transition from $iOFS$ to IFS.

## Modeling of EAAT1s

Homology models for human EAAT1 (K42 to S498) in the $iOFS$ (EAAT1_$iOFS$) and $iChS$ (EAAT1_$iChS$) states were constructed using MODELLER based on an asymmetric Glt$_{Ph}$ crystal structure (*Verdon and Boudker, 2012*) and our in-silico resolved channeling intermediate of the Glt$_{Ph}$. We adopted the multiple sequence alignment of Glt$_{Ph}$, glutamate and neuronal amino acid transporters reported earlier (*Yernool et al., 2004*). The segment connecting TM4b and TM4c (E184 to S237 in EAAT1) was not included in the analysis due to the lack of a homologous portion in Glt$_{Ph}$. D400, D476 and D487 were protonated based on pKa calculations using PROPKA (*Li et al., 2005*). Then the transmembrane domain of the EAAT1 trimer was inserted into the center of a pre-equilibrated POPC membrane, following the same approach for Glt$_{Ph}$. For each system, two 100 ns MD simulations were performed, in which external electrostatic fields of 0.1 kcal/(mol.Å.e) (i.e. 150 mV across the membrane) and −0.1 kcal/(mol.Å.e) were applied, respectively. Similarly, R477C EAAT1 mutant in the $iChS$ was constructed and two 50 ns MD simulations were performed, in which external electrostatic fields of 0.1 kcal/(mol.Å.e) and −0.1 kcal/(mol.Å.e) were applied, respectively.

## Trajectory analysis

Radii of the potential chloride channels were measured using the HOLE program (*Smart et al., 1996*). VMD (*Humphrey et al., 1996*) with scripts adopted from VMD script library (http://www.ks.uiuc.edu/Research/vmd/script_library) was used to analyze the time evolution of structures, such as the backbone RMSD, the average pore radii and residue SASA. The RMSDs refer to the structurally resolved regions of subunit C. The conformation state was assessed by z-coordinate distances of the mass centers of HP2 (V335-V370), HP1 (P258- K290) related to S65 that remained minor movement during transitions.

# Acknowledgement

We gratefully acknowledge support from NIH grants P41GM103712, P30DA035778 and 5R01GM099738 to IB and ZIAMH002946 to SGA. We thank Dr. Jennie Garcia-Olivares for thoughtful comments on the manuscript.

## Additional information

### Funding

| Funder | Grant reference number | Author |
|---|---|---|
| National Institutes of Health | P41GM103712 | Ivet Bahar |
| National Institutes of Health | P30DA035778 | Ivet Bahar |
| National Institutes of Health | 5R01GM099738 | Ivet Bahar |
| National Institutes of Health | MH002946 | Susan G Amara |

The funders had no role in study design, data collection and interpretation, or the decision to submit the work for publication.

### Author contributions

MHC, Formal analysis, Validation, Investigation, Visualization, Methodology, Writing—original draft, Writing—review and editing, Designed the experiments, performed computational study and analysis, and wrote the paper; DT-S, Formal analysis, Validation, Investigation, Methodology, Writing—original draft, Writing—review and editing, Designed the experiments, performed experimental study and analysis, and wrote the paper; ADG-S, Formal analysis, Investigation, Performed experimental study and analysis; SGA, Supervision, Validation, Methodology, Writing—original draft, Writing—review and editing, Designed the experiments and wrote the paper; IB, Formal analysis, Supervision, Validation, Methodology, Writing—original draft, Writing—review and editing, Designed the experiments, performed computational study and analysis, and wrote the paper

### Author ORCIDs

Mary Hongying Cheng, http://orcid.org/0000-0001-5833-8221
Ivet Bahar, http://orcid.org/0000-0001-9959-4176

## Additional files

### Supplementary files

• Supplementary file 1. Summary of MD simulations.

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
