## [Decision Letter]

[Editors’ note: this article was originally rejected after discussions between the reviewers, but accepted after the authors appealed against the decision.]

Thank you for submitting your work entitled "Substrate transport and anion permeation proceed through distinct pathways in glutamate transporters" for consideration by *eLife*. Your article has been reviewed by three peer reviewers, one of whom is a member of our Board of Reviewing Editors and the evaluation has been overseen by a Senior Editor. The following individuals involved in review of your submission have agreed to reveal their identity: Christof Grewer (Reviewer #2).

Our decision has been reached after consultation between the reviewers. Based on these discussions and the individual reviews below, we regret to inform you that your work will not be considered further for publication in *eLife*.

Using molecular dynamics simulations combined with biochemical and functional assays, this study explores the structure and mechanism of the chloride permeation channel of secondary-active glutamate transporters. While the findings are largely consistent with the earlier findings, they fall short of providing significant new insights. It appears that, in a close look, the identified transporter conformation and the channel for chloride permeation do not differ substantially from that identified and characterized by Machtens et al., 2015.

Reviewer #1:

Cheng et al. report an investigation of the mechanism by chloride ions permeate across secondary-active glutamate transporters, through passive diffusion. To this end, the authors combine molecular dynamics simulations and biochemical and functional studies. The former are primarily focused on Glt_Ph_, a prokaryotic transporter, while the experimental assays are carried out for human EAAT1, expressed in oocytes.

The most apparent shortcoming of this study is that it is unclear how it adds to the published body of work on this subject – particularly recent work, e.g. Machtens et al., 2015. The principal conclusion in that study is essentially the same as that proposed here, namely that an intermediate conformation between the OF and IF states is such that an anion-selective conducting pore opens up at the interface between the transport and trimerization domains; as far as I can discern by comparing the figures in these two articles, even the proposed pathway for chloride is highly similar.

It could be argued that independent verification of that result would be of interest. For example, a reader of Machtens et al. might be concerned by the fact that the conducting intermediates hypothesized that study originate from computer simulations in which the transporter is subject to transmembrane voltages of up 1.6 V. Thus, a re-evaluation of those results with more sophisticated or systematic approaches would be well justified. This is not the case here, however. The computational work in Cheng. et al. is of subpar quality, from a methodological standpoint, and appears to follow a design that is largely arbitrary. For example, the authors use so-called targeted MD simulations to generate conformational intermediates between the so-called iOFS and IF states. In this approach, the RMSD of the protein backbone structure, relative to a target, is gradually reduced to a value close to zero. This approach might seem reasonable, except these TMD trajectories are 10 ns long – i.e. clearly out of equilibrium. In that timescale, there simply is no chance that the side-chain structure, which is, after all, what controls the backbone conformation, will be minimally realistic. In my opinion, therefore, it is improbable that the resulting structural intermediates fall close to a minimum free-energy pathway between the two states considered – except in a purely qualitative sense. This issue aside, why exactly are the authors assuming here that the Cl^-^ conducting state is an intermediate between iOF and IF conformations? As mentioned, the work seems to examine the same concepts put forward elsewhere. The authors should also clarify why they reportedly carry out 6 of these non-equilibrium trajectories, but apparently consider only one configuration of run #1 for further analysis. Why this precise configuration? What is the significance of these multiple calculations?

The subsequent steps of the computational protocol are equally confusing and seemingly ad hoc. Why are multiple simulations of the selected intermediate carried out with and without voltage applied? In which way does this procedure lead to the "refinement" of the structure of this intermediate? What is the metric that enables the authors to consider this structural intermediate is "refined"? And why, again, is the endpoint of only one of these multiple simulations considered for the analysis of ion permeation through PMF calculations? Are the authors using two different methods to compute these PMF profiles (Metadynamics and ABF), as the supplementary information seems to suggest? Why would that be necessary? What is the statistical error in the calculated PMF profiles? And importantly, what is the Cl^-^ conductance associated with the calculated PMF profile, and does it resemble the experimental value? Finally, why are these PMF profiles (for Cl^-^ and Na^+^) computed in the absence of one of the 3 Na^+^ ions bound to the structure?

I don't have major concerns in regard to the experimental data. However, I fail to recognize in which way this data specifically supports or confirms the computational work presented here, except on a very qualitative sense. As far as the question of Cl^-^ channeling is concerned, the main experimental result is that MTS modifications of Cys substitutions at EAAT1 positions equivalent to Val51 and Leu212 Glt_Ph_ reduce the rate of ion permeation, but not transport. The structure of Glt_Ph_ shows these residues cluster together at the above mentioned interface, and clearly exposed to the solvent on the extracellular side. This data therefore helps to confirm the previously proposed notion that Cl^-^ diffuses across this interface – it also confirms the data for V51 mutants in Machtens et al., 2015. What I fail to recognize is (a) what is distinct about the computational predictions put forward here, and (b) in which way the experimental data obtained in this study specifically validates those predictions.

Reviewer #2:

This is very nice and comprehensive work that reveals novel information on the mechanims of the anion conductance associated with glutamate transporters. The combination evidence from molecular dynamics simulations supported by experimental data is compelling. I have no general concerns, but would like to ask the authors to include citations of the work of Tao and Grewer regarding the location of the Na3 binding site, which is used in this manuscript for the MD simulations.

[Editors’ note: the author responses to the first round of peer review follow.]

---

## [Author Response]

*Reviewer #1:*

*Cheng et al. report an investigation of the mechanism by chloride ions permeate across secondary-active glutamate transporters, through passive diffusion. To this end, the authors combine molecular dynamics simulations and biochemical and functional studies. The former are primarily focused on* Glt_Ph_*, a prokaryotic transporter, while the experimental assays are carried out for human EAAT1, expressed in oocytes.*

The Reviewer stated that our simulations were primarily focused on the prokaryotic transporter. This is incorrect. We performed detailed computations for the human EAAT1 as well, which have been experimentally investigated. We have repeated the majority of simulations for both the archaeal (Glt_Ph_) and human (EAAT1) transporters.

*The most apparent shortcoming of this study is that it is unclear how it adds to the published body of work on this subject – particularly recent work, e.g. Machtens et al., 2015. The principal conclusion in that study is essentially the same as that proposed here, namely that an intermediate conformation between the OF and IF states is such that an anion-selective conducting pore opens up at the interface between the transport and trimerization domains; as far as I can discern by comparing the figures in these two articles, even the proposed pathway for chloride is highly similar.*

We agree that our proposed anion permeation pathway is similar to the one proposed by Machtens et al. But it should be noted that (Bahar, 2014) there was an ambiguity (if not disagreement) in the literature on this topic, and now our study clearly supports the results of Machtens et al; and (2) we also provide new data beyond what has been reported. Our work demonstrates that the opening of the channel proceeds through a cooperative dislocation of a flexible-wall that physically separates the anion permeation pathway and the substrate translocation site. This new concept was notreported in earlier works. The role of this flexible wall is demonstrated by modifications that significantly affected anion conduction without interfering with substrate transport. Our data indicated that substrate transport and anion permeation proceed through these two mutually exclusive pathways, by virtue of the conformational flexibility of the wall which may obstruct one pathway while opening the other, and *vice versa*. We further showed that this mechanism is shared by both the archaeal and mammalian transporters.

*It could be argued that independent verification of that result would be of interest. For example, a reader of Machtens et al. might be concerned by the fact that the conducting intermediates hypothesized that study originate from computer simulations in which the transporter is subject to transmembrane voltages of up 1.6 V. Thus, a re-evaluation of those results with more sophisticated or systematic approaches would be well justified. This is not the case here, however.*

Transmembrane voltages of up 1.6 V and the usage of 1M NaCl (in Machtens et al) are too far off from reality. Glt_Ph_ has over 150 charged residues and EAAT1, over 200 charged ones. Application of a high voltage (1.6 V) would significantly alter the side chain orientations of these charged residues, thus generating structures that would depart from those stabilized under physiological voltages used in experiments. Indeed, we have employed a more sophisticated computational methodology to avoid high voltages (up to 300 mV) and high sodium concentrations (0.15 M), which permitted us to identify the chloride permeation pathways in both Glt_Ph_ and EAAT1. The methodology we implemented here should find valuable applications in other membrane proteins too.

*The computational work in Cheng. et al. is of subpar quality, from a methodological standpoint, and appears to follow a design that is largely arbitrary. For example, the authors use so-called targeted MD simulations to generate conformational intermediates between the so-called iOFS and IF states. In this approach, the RMSD of the protein backbone structure, relative to a target, is gradually reduced to a value close to zero. This approach might seem reasonable, except these TMD trajectories are 10 ns long – i.e. clearly out of equilibrium. In that timescale, there simply is no chance that the side-chain structure, which is, after all, what controls the backbone conformation, will be minimally realistic. In my opinion, therefore, it is improbable that the resulting structural intermediates fall close to a minimum free-energy pathway between the two states considered – except in a purely qualitative sense.*

We respectfully disagree. Our simulations use well-established methods of molecular computational biology. However, we take responsibility for not being clear enough and induce what we believe is a misinterpretation by reviewer #1. We are now explaining in more details. Multiscale methodologies combine conventional MD (cMD) simulations for visualizing local events, targeted MD (tMD) for directing the motion to undergo functional transitions, and/or accelerated MD (aMD) for simulating processes of biological interest otherwise not accessible by conventional MD. As such they present useful tools for exploring coupled global and local events. We adopted such an approach here: we performed a series of cMD simulations guided by short tMD runs (see [Supplementary-material SD1-data]). It is important to note that tMD runs ought to be short (e.g. 10 ns), their goal is simply to enable the sampling of conformers that are otherwise beyond the reach of cMD. If these runs are conducted for longer (than for example 10 ns) durations, they may artificially drive the molecule toward unrealistic conformations – especially if the force constants associated with the targeted harmonic constraints are too large. To avoid such situations, three important requirements are (i) to apply soft forces on the backbone (only), and (ii) to conduct these runs for short durations (as we did), and (iii) to follow tMD runs by long cMD simulations that allow the biomolecule to relax and sample energetically favorable conformers (note that our tMD runs were succeeded by cMD runs of 100s of nanoseconds – which the Reviewer has apparently overlooked). So, tMD runs are not expected to reach equilibrium conformers for the side chains (nor the backbone). The succeeding cMD runs do. The generation of short tMD trajectories followed by long cMD is a well-established protocol (see for example Schulz R et al. 2010: PLoS Comput Biol 6: e1000806. Shaikh and Tajkhorshid, 2010 and our previous work, Cheng and Bahar, 2013 and 2014.

*This issue aside, why exactly are the authors assuming here that the Cl^-^ conducting state is an intermediate between iOF and IF conformations?*

This assumption (or assessment) was based on two observations consistently made in independent runs. First, in all 6 independent runs, we consistently observed a water channel to form (or a hydrated conformer to stabilize) when the RMSD reached 3.5-4.0 Å with respect to the *i*OFS, and the transporter became dehydrated once it approached the IFS (e.g. RMSD > 5.5 Å). It was clear that an open form that could conduct ion facilitated by the influx of intracellular water would be this intermediate hydrated conformer that exhibited a stable water channel, with structural characteristics distinct from both the iOF and IF states (see Figure 7).

*As mentioned, the work seems to examine the same concepts put forward elsewhere. The authors should also clarify why they reportedly carry out 6 of these non-equilibrium trajectories, but apparently consider only one configuration of run #1 for further analysis. Why this precise configuration? What is the significance of these multiple calculations?*

As we mentioned, in all six simulations, we consistently observed the intermediate water channel state. We simply reported one of these equilibrated conformers for illustrative purposes, and we are happy to provide in the supplementary material those obtained in the individual runs.

*The subsequent steps of the computational protocol are equally confusing and seemingly ad hoc. Why are multiple simulations of the selected intermediate carried out with and without voltage applied?*

It is well-known that the chloride channels in the glutamate transporters are voltage-gated. So we performed multiple simulations of selected intermediate with and without voltage applied.

*In which way does this procedure lead to the "refinement" of the structure of this intermediate? What is the metric that enables the authors to consider this structural intermediate is "refined"?*

The refined or stabilized channel was confirmed by a stabilized RMSD from the initial conformer, small fluctuations in the channel pore and continuous water occupancy over the time course of cMD runs.

*And why, again, is the endpoint of only one of these multiple simulations considered for the analysis of ion permeation through PMF calculations?*

In multiple runs, the water channel conformers were overall similar, if not identical. PMF calculations are usually time-consuming (up to several 100 ns). We used one as the initial conformer, but the PMF calculation averages over all the configurations within the sampling time scale (~100 ns). Note that our approach to estimate PMF differs from that used by Machtens et al., who used continuum approach (Poisson-Boltzmann) on a static conformer.

*Are the authors using two different methods to compute these PMF profiles (Metadynamics and ABF), as the supplementary information seems to suggest? Why would that be necessary? What is the statistical error in the calculated PMF profiles? And importantly, what is the Cl^-^ conductance associated with the calculated PMF profile, and does it resemble the experimental value? Finally, why are these PMF profiles (for Cl^-^ and Na^+^) computed in the absence of one of the 3 Na^+^ ions bound to the structure?*

We used both metadynamics and ABF to estimate the free energy profiles along the putative pathways, as a way of consolidating the results. These two methods complement each other. Metadynamics has the advantage of identifying curved permation pathways and estimating the free energy profiles with less computational effort, albeit at lower accuracy compared to other methods (see Zhang and Voth (2011) J. Chem. Theory Comput 7, 2277-2283) like ABF. We used metadynamics for a first, quick assessment of putative chloride channeling pathways. Then the one with the lowest energy barrier was reevaluated using ABF to achieve higher accuracy. The standard error in the calculated PMF profiles are presented in standard way shown in Figure 5, with the *light* and *cyan* shaded area corresponding to sodium and chloride ion translocation, respectively. Note that the purpose of PMF calculations (Figure 5) was to identify the ion selectivity region and the gating residues. Computation of chloride conductance is beyond the scope of the present study. Finally, the binding site of the third Na^+^ ion is quite far (> 20 Å) from the permeation pathway (see Figure 2—figure supplement 1), and as such it would have a minimal effect, if any, on the computed free energies.

*I don't have major concerns in regard to the experimental data. However, I fail to recognize in which way this data specifically supports or confirms the computational work presented here, except on a very qualitative sense. As far as the question of Cl^-^ channeling is concerned, the main experimental result is that MTS modifications of Cys substitutions at EAAT1 positions equivalent to Val51 and Leu212* Glt_Ph_*reduce the rate of ion permeation, but not transport. The structure of* Glt_Ph_*shows these residues cluster together at the above mentioned interface, and clearly exposed to the solvent on the extracellular side. This data therefore helps to confirm the previously proposed notion that Cl^-^ diffuses across this interface – it also confirms the data for V51 mutants in Machtens et al., 2015. What I fail to recognize is (a) what is distinct about the computational predictions put forward here, and (b) in which way the experimental data obtained in this study specifically validates those predictions.*

The experimental data support the computational predictions by confirming that three of the residues that appear to be critical to form the anion permeation cavity in both Glt_Ph_ and hEAAT1 (V51/M89, L212/L296 and F50/L88) all showed accessibility to MTS reagents; and modification with these reagents significantly affected anion permeation, which is a classical approach used by many others to identify pore forming residues. The facts that in the SASA calculations for hEAAT1, (i) these three residues are the only ones that showed a high SASA value in the predicted channeling conformation (iChS), and (ii) at the same time their SASA was significantly different than those values obtained for the same residues in the inward-facing (IFS) or the intermediate outward-facing conformation (iOFS), further shows the significance of computational predictions, which were confirmed by experiments.

Moreover, the fact that modifications of two of these residues altered anion permeation without affecting substrate transport, support the computational prediction that the two pathways are physically separated.

In addition, the substitution of L88 for an arginine, resulting in altered permeability ratios for all permeant anions, further supports the computational prediction that this residue is deep into the pore, probably interacting with the permeant anion and even the selectivity filter.

Furthermore, the experimental data in Figure 8 demonstrate that when the carriers are exposed to conditions that favor either the outward- (OFS) or the inward-facing state (IFS), those residues are less accessible, or even not accessible at all. These data confirm the novel finding proposed by our computational predictions that only intermediate conformations allow opening of the anion permeation pathway and therefore that anion channel opening is state-dependent.

*Reviewer #2:*

*This is very nice and comprehensive work that reveals novel information on the mechanims of the anion conductance associated with glutamate transporters. The combination evidence from molecular dynamics simulations supported by experimental data is compelling. I have no general concerns, but would like to ask the authors to include citations of the work of Tao and Grewer regarding the location of the Na3 binding site, whcih is used in this manuscript for the MD simulations.*

We are grateful to Dr. Grewer for supportive comments. We agree that the work of Tao and Grewer should have been cited, and we are happy to add this citation to the manuscript.